# Evolution Model and Simulation Study of the Public Risk Perception of COVID-19

**DOI:** 10.3390/ijerph191811581

**Published:** 2022-09-14

**Authors:** Ao Zhang, Hao Yang, Zhenlei Tian, Shuning Tong

**Affiliations:** 1School of Engineering and Technology, China University of Geosciences (Beijing), Beijing 100083, China; 2Emergency Management Department of Xinjiang Uygur Autonomous Region, Urumqi 830011, China

**Keywords:** risk perception, COVID-19, limited memory theory, risk information, evolution model

## Abstract

The evolution of the public perception of the risk in public health emergencies is closely related to risk response behavior. There are few systematic explanations and empirical studies on how the individual receiving the risk information affects the change in the individual risk perception through internal mechanisms in the context of COVID-19. Based on the understanding of the existing research, this paper constructs the evolution model of the public risk perception level based on the limited memory theory and a simulation analysis is performed. The results are as follows: memory rate, association rate, information reception and information stimulation in a single period of time have significant indigenous effects on the risk perception; when the amount of information received and the information stimulus remain unchanged, the public’s risk perception follows a monotonic upward trend, but there is an upper limit function, and the upper limit is determined by the memory rate and association rate, and the influence of the association rate is higher than that of the memory rate; When the amount of information received and the information stimulus changes, the public’s risk perception will also change, and there is a lag effect, which is determined by the memory rate. The impact of the acceptance of the information on the risk perception is greater than that of the information stimulus.

## 1. Introduction

It has been more than two years since the World Health Organization declared COVID-19 “an international public health emergency”. In each stage of the COVID-19 epidemic, social media has become the main way for the public to obtain external information. Especially during the most serious part of the epidemic, that is, in early 2020, when the number of searches associated with the epidemic reached the hundreds of millions. Social media can be used not only to provide public access to real-time epidemic news, but also to address the shortage of PPE (Personal Protective Equipment) [1]. Various sources of information impact people’s awareness of COVID-19. Information such as epidemic situation notifications, government preventions and control measures, disease preventions and control knowledge as well as folk rumors make people more alert. This study focusses on the various types of epidemic information and the perception risks received by individuals during public health emergencies.

Risk perception has its basis in psychology. It is the individual’s subjective feeling and understanding of the various objective risks found in the external environment. Currently, the definition of risk perception has not yet formed a unified conclusion [2]. At present, the most authoritative definition in academia was put forward by Paul Slovic [3] in 1987, who believed risk perception to be an intuitive judgment of a target risk by individuals or organizations in the context of limited information and uncertainty. Since the concept of risk perception was first proposed by Paul Slovic [3], risk perception has been widely used in areas of public crises such as natural meteorological disasters and sudden infectious diseases. Risk information is found to play an important role in the evolution of risk perception because most people lack the intuitive experience and feelings concerning risk [4]. Dennis and Lori’s study [5] suggests that risk information contributes to the public’s risk perception. For an individual perception, the characteristics of risk information, such as information sources, content, distribution channels, reporting forms, information density and other factors, will even exceed the impact of the individual characteristics concerning risk perception. In terms of information sources and channels, Lindell and Perry [6] proposed the classic six-step communication model (source, channel, information, recipient, influence and feedback), which mentioned that the information source plays a basic role in the process of information dissemination. Chung [7] noted that in most cases, risk perceptions and the related background knowledge are not acquired through direct personal experience, and that different sources and channels of information play a key role in the process of the public’s perception of a risk. In terms of information content, Kan Shi, Hongxia Fan et al. [8] classified the SARS-related information into four categories: positive and negative information, risk-related information and risk prevention measures, and put forward the main factors of a risk assessment, namely, disease information, cure information, information closely related to the risk assessment and government preventive measures. It is understood that negative information is more likely to negatively affect an individual’s cognition, while positive information reduces people’s level of risk cognition, risk-related information can increase risk perception, and the program measures information can reduce risk perception, compared with positive information. Negative information is more sensitive to public perception. Finally, for other information characteristics, Klemm, Hartmann and Das [9] pointed out that the continuity, reliability and accuracy of information released by the media are important factors affecting the public risk perception. Emotional news forms an increasing public awareness concerning the severity of the disease.

In the research field of risk information and risk perception, most of the studies focus on the exploration and verification of the influencing factors of risk perception. They mainly use a survey as the research method to qualitatively describe and summarize the impact of risk information on risk perception, while there are few studies on the evolution process of risk perception from the perspective of time. The study found that after processing the received risk information, people formed their own understanding and cognition of the risk and stored it in their memory [10]. Therefore, many scholars have also used memory theory to study the relationship between the public’s acceptance of risk information and the change of the risk perception level. For example, Jiuchang, Fei and Dingtao [10] established the mathematical model for risk information and risk perception in natural disasters by using the Ebbinghaus forgetting curve and carried out simulations and analyses according to the release model of three kinds of disaster information. For other information characteristics, it was pointed out that the continuity, reliability and accuracy of the media information are important factors affecting the public risk perception. Emotional news forms an increasing public awareness of the severity of the disease.

From the perspective of academic research, the impact of the COVID-19 epidemic is far beyond the scope of biomedicine. It is a crisis that requires comprehensive interdisciplinary methods and collective scientific efforts to help understand and mitigate its security impact [11]. Currently, there is no research exploring the level of the public risk perception from the perspective of individual memories. Therefore, this study attempts to explore the following questions from the perspective of individual memories, based on the limited memory theory, with the individual as the research subject and the COVID-19 event as the research case:In the face of a sudden epidemic of infectious diseases with a strong uncertainty and high risk, how do people understand and deal with the epidemic?In each stage of an epidemic evolution, what changes will occur in the composition and quantity of the different types of epidemic information, and the corresponding changes in the level of risk perception?Whether or not the way different people experience the same information acceptance process will produce different evolution processes of risk perception.

This study attempts to explain the dynamic evolution mechanism of the individual risk perception level in the process of public health emergencies by studying the above issues.

## 2. Theoretical Basis

### 2.1. Limited Memory Theory and the Individual Memory Model

Since the German psychologist Hermann Ebbinghaus published his experimental report, the field of individual memory has become a hot topic in psychological experimental research. In 1969, Shiffrin and Atkinson proposed a three-stage processing model of memory information [12], which argued that as a complete memory system, it included sensory memory, short term memory and long term memory. Once the external information enters the memory system, it goes through these three memory structures for processing, and through the specific function of each memory structure, the memory information also goes through three stages, from low to high. The three memory structures in the three-stage processing model of memory information are independent. Since then, with the continuous development of the individual memory theory, some scholars have explored the rules and characteristics of memory through experiments. The limited memory theory is an important part of the individual memory theory. In other words, the human memory is limited and it is impossible to accurately process all of the information received. It tends to focus on the most important fragments and ignore their respective parts. Taking the new coronavirus as an example, it is assumed that when receiving the relevant information, the public may focus on the number of confirmed cases, the location of the incident and other eye-catching fragments, while ignoring their information.

Sendhil [13] summed up the association effect of memory (the association effect of memory, also known as the logical miscalculation effect, refers to the association as the psychological process of one thing to think of another thing. This means that if we are remembering the same thing, the second time will be more effective than the first. For example, when we remember a phone number, the more times we repeat it, the more firmly we will remember it.) and the retelling effect (the retelling effect of memory refers to the memory of an event that can promote its subsequent memories), combined with the recency effect proposed by the American psychologist A. Ladins (the recency effect of memory is the phenomenon whereby people remember a series of things in the last part of the transaction better than the middle part of the transaction). For the first time, the psychological effects of three kinds of memory are applied to the field of economics through mathematical modeling, and the economic model of individual memory is established.

Since then, the recency effect, association effect and the retelling effect and the establishment of individual memory models, based on these three theories, have been widely used in various fields. For example, Sarafidis [14] proposed how an agent or broker should release information to influence the judge’s memory and improved the memory model based on three effects and Jiuchang, Fei, and Dingtao [10] constructed a public risk perception memory model to explore the changes of public perception in different news reporting modes.

### 2.2. Crisis Life Cycle and the Management Model

In the face of sudden crisis events, the measures taken to quickly and efficiently prevent the spread of the crisis as well as the different ways to solve the crisis, have always been important research topics for scholars. Equally, the exploration of the crisis life cycle helps to scientifically divide the development stages of the crisis events in order to better clarify the role of individual memory and the risk information cognition in the different stages and their corresponding changes. If we can scientifically divide the development stages of crisis events, we can take targeted emergency measures, distinguish responsibilities and strengthen the mechanisms or procedures in the corresponding periods, to implement a faster and more professional emergency response. Therefore, scholars have continued to explore the cycle division of the crisis and crisis management in order to take the appropriate measures, according to the cyclical characteristics of the sudden crisis events more reasonably. Qunying Xiao and Huijun Liu [15], based on their analysis of the life cycle of the SARS epidemic, combined with the epidemic data and the development trends, they also divided the life cycle model into five stages and set the corresponding critical standard system, as shown in Table 1.

Jie Guo, Lichang Yang and Zixu Sun [16] believe that crisis management and crisis emergencies are not independent of each other. It is only by fully understanding the development law of the crisis emergencies that we can grasp the periodicity of the crisis management more comprehensively. Once we have considered the lag effect (the emergence of each stage node of the emergency is earlier than that of each stage node of the crisis management), a hyperbolic model of the emergency crisis management in time dimension is established, according to the characteristics of the time period. Based on the description and understanding of the different stages, the classification criteria are summarized as shown in Table 2.

Of the two listed criteria for dividing the life cycle of an epidemic, the first is more practical in that it gives a specific division index. However, because the time period of the author’s study is in the early stages of the solution period, there is no domestic epidemic response that is good. The foreign epidemic response is poor, the focus of the work to prevent and control the epidemic has shifted to a situation calling for an overseas prevention and control, and this results in the critical standard of the solution period and the recovery period cannot be reached. The domestic epidemic has been in its recovery period which is not consistent with the fact. The second criterion is more theoretically inclined. It interprets the cycle from the characteristics of the crisis events and presents the characteristics of each period. However, it is necessary to select the indicators of the actual events again. Moreover, the standard of the two divisions above has its own advantages and disadvantages, so this study will combine the two in order to construct a more scientific and practical classification criterion, to reselect the indicators of the actual events under study.

## 3. Methods

Based on the limited memory theory, this paper discusses how the risk perceptions of individuals can change with the reception of the epidemic information during the COVID-19 outbreak (27 December 2019 to 2 May 2020) from the perspective of individual memory, in order to better understand the impact of the release of the epidemic information on the individual coping behaviors. Combining the near-factor effect, the retelling effect, and the association effect in memory theory, the evolution model of the epidemic risk perception is constructed through mathematical modeling, and the influence mechanism of the epidemic information release mode and individual memory characteristics on risk perception is discussed through a simulation.

### 3.1. Construction of a Public Risk Perception Model of COVID-19

#### 3.1.1. Scenario Introduction

From the early days of the pandemic, people have been filtering new information through different cultural and ideological lenses. For the public, following the outbreak of COVID-19, on the one hand, the public will personally feel the impact of the epidemic on their own lives, health and safety, as well as the various social and economic aspects. Even the less media-focused public can feel the changes (masking, quarantines, etc.) implemented in everyone’s daily lives during the pandemic. On the other hand, the public will also receive a large amount of epidemic information from the social media. The growing flow of information and the rapidly escalating situation has increased the visibility of new crown pneumonia in the media and on social media [17]. Whether or not personal experience or information received from the network will affect the public’s memory and cognition of the new corona pneumonia, and the individual characteristics of the receiver, will determine the formation of this cognition. Due to the persistence of the cognitive process and the Ebbinghaus forgetting curve, the public perception of the epidemic will change over time. For example, the public perception of the epidemic will change with the situation of the epidemic itself. However, when the severe events in the epidemic occur again, the spread of the epidemic information will once again affect the public perception. For example, the sudden increase in the number of confirmed cases will make the public perception of the epidemic rise again, and the success of vaccine testing will reduce the public perception. Therefore, the public perception of the epidemic is a dynamic process, and the memory effect of time will always exist, while the emergence of the severe events in the epidemic will have a new impact on the public perception. For example, as the virus species of an outbreak mutates over time, the measures we have to take in response to the outbreak, such as quarantine, masking, and social distancing, will change accordingly. As the epidemic process evolves, the previous epidemic prevention messages are slowly forgotten by the public, and the new messages then have a new impact. The public perception of the epidemic is the basis for the public risk response behavior in the epidemic situation. When the public perception reaches a certain level, it will promote them to take a relevant response behavior.

Following the outbreak of the general epidemic, the media reports and spreads a large amount of epidemic information on the same day. If each day of the epidemic is regarded as an event composed of the information about the epidemic received every day, then the period of an event is regarded as a period of time. Once the public has experienced the first period, the public feels the change of the epidemic through the media and changes its perception of the risk. With the passage of time, the public’s memory of the epidemic gradually fades, and with the change of the daily epidemic data and the occurrence of severe events, the number and content of the epidemic-related information and reports will also change. This leads to changes in the individual risk perception after each epidemic event.

#### 3.1.2. Research Assumptions

Based on the previous review of the studies related to the risk perception influencing factors, it can be seen that there are four broad types of influencing factors for public risk perception: individual public characteristics and experiences, risk nature characteristics, risk information characteristics and time.

The characteristics of the individual members of the public include gender, age, occupation, education level and family cultural background. The differences in life experiences and risk attitudes caused by the characteristics of the individual members of the public are the most direct reasons for the differences in the individual risk perceptions. Combined with the individual memory theory, these individual characteristics tend to influence the individual’s memory and association rates. In addition to the characteristics of the individual, the characteristics of the risk itself, as the object of perception, can also have an impact on the risk perception of the risk. In this study, the study context was set during the COVID-19 outbreak (27 December 2019 to 2 May 2020), and the characteristics of the risk itself can be identified as the characteristics of the COVID-19 epidemic, so its influence is stable and not discussed in the model.

In addition, for the individual perception, the influence of the risk information characteristics, such as the information source, content, distribution channel, reporting format and information density, can even exceed the influence of the individual characteristics on the risk perception. This study will characterize the complexity of the information in an epidemic through two variables: the information receiving amount for each period and the amount of stimulation of the single information for each period.

Finally, time is an important dimension for the study of the evolutionary process of the public risk perception. First, the public will continuously engage in the act of collecting and accumulating information in order to avoid risks and reduce risk perceptions. Second, the information capacity of the public is constant [18], which leads the public to continuously update the information they have, and the utility of the earliest information will continuously diminish in comparison due to the forgetting effect of the memory. The passage of time is a necessary condition for the above process, so time is also an important influence on the risk perception. The specific research hypotheses and variables are presented as follows.

Epidemic information release assumption

Based on the limited memory theory and related models, when we study the perception of the epidemic risk from the perspective of the individual memory, we need to simplify the individual as a complete information audience. The audience of the information represents the changes in the epidemic in each time period, the individual can receive relevant information and the amount of epidemic information received by the public after the *t* time period is *N_t_*, then the information received by the public from the first time period to the t time period is set D = {*N*_1_, *N*_2_, …, *N_t_*}. Next, we examine the changing process of the public perception of the epidemic in each time period separately. To simplify the model, we assume that each individual receives each epidemic information at the same time interval and set it to $\frac{1}{N_{t}}$.

Considering the different types and contents of information in each time period, we assume that the stimulus amount of Article *i* information to the perception of the individual risk of the epidemic in each time period is *S_i_*. To reflect the complexity of the information in the epidemic and to ensure that the model is easy to deduce, we believe that each information brings the same perception to the individual in each time period, but in different time periods, a single information is different from the perceived stimulus, and the perceived stimulus is determined by the type of information received by the individual. According to the relevant studies, the increase in the proportion of the negative information and the epidemic information itself will improve the perception of the individual epidemic risk, while the increase in the proportion of the positive information and the epidemic prevention and control measures will reduce the perception of the individual epidemic risk [8]. In addition, disinformation is indeed a feature that affects the perception of the COVID-19, but mis/false information is also part of that perception, as is the related explanatory and clarifying information. This study argues that the effect of fake news on the level of the risk perception is also dependent on the amount of the single message stimuli, the number of messages and the positive and negative nature of the messages, which is the same as true messages. We believe that the influence mechanism of the positive messages will not be confused with the influence mechanism of negative messages due to the change in the truthfulness of the messages.

2.Memory storage assumption

Based on the assumption of the memory parameters [14] by the limited memory theory, the Ebbinghaus forgetting curve, and related models, the recency effect is introduced into the model. The recency effect indicates that memory fades over time, so the probability that individuals forget information due to an interference after receiving the information ranges from 0 to 1. This can be explained by the Ebbinghaus memory curve. It believes that when an individual accepts the information, its memory will show an exponential decay over time, that is, M(*t*) =$\;e^{-\rho_{t}}$, where M(*t*) denotes the individual’s memory of the information at time *t*, and *ρ* denotes the individual’s memory rate. Then, combined with the influence of the association effect, the new information will trigger memories of past information. For example, reading an article depicting a good image of a politician in a newspaper will arouse other good memories associated with him. Moreover, the stimulation of the new information recall on the perception also depends on the time interval between the information and all of the previous information, and each person has an association rate k, which is related to individual characteristics and events.

### 3.2. Simulation of the Public Risk Perception Model of COVID-19

#### 3.2.1. Simulation Software

In this paper, we chose MATLAB software for the simulation. Unlike other programming software, MATLAB uses a matrix operation instead of a circular operation, which improves the operation speed and completes the collective processing of the parameter set in this paper. Additionally, MATLAB has a convenient data visualization function that can generate data images immediately after the data processing is complete.

#### 3.2.2. Simulation Method and Process

Finally, according to the research assumptions under Section 3.1.2, the risk perception evolution model of the individuals in the epidemic situation is obtained in the form of the first formula (1) and the recursive formula (2) [19].
(1)$$y_{N_{1}}=S_{1}\cdot \left[\frac{\rho^{\frac{1}{N_{1}}+1}}{\left(\rho^{\frac{1}{N_{1}}}-k^{\frac{1}{N_{1}}}\right)\left(\rho^{\frac{1}{N_{1}}}-1\right)}-\frac{k^{\frac{1}{N_{1}}+1}}{\left(\rho^{\frac{1}{N_{1}}}-k^{\frac{1}{N_{1}}}\right)\left(k^{\frac{1}{N_{1}}}-1\right)}+\frac{1}{\left(\rho^{\frac{1}{N_{1}}}-1\right)\left(k^{\frac{1}{N_{1}}}-1\right)}\right]$$
(2)$$y_{N_{t}}=\rho \cdot y_{N_{t-1}}+\left[\frac{\rho -k}{\rho^{\frac{1}{N_{t}}}-k^{\frac{1}{N_{t}}}}\cdot 1-k\cdot \left(\;\sum_{i=1}^{t}\frac{k^{\frac{1}{N_{i}}}}{1-k^{\frac{1}{N_{i}}}}\cdot k^{t-i}\right)+\frac{\rho^{\frac{1}{N_{t}}+1}}{\left(\rho^{\frac{1}{N_{t}}}-k^{\frac{1}{N_{t}}}\right)\left(\rho^{\frac{1}{N_{t}}}-1\right)}-\frac{k^{\frac{1}{N_{t}}+1}}{\left(\rho^{\frac{1}{N_{t}}}-k^{\frac{1}{N_{t}}}\right)\left(k^{\frac{1}{N_{t}}}-1\right)}+\frac{1}{\left(\rho^{\frac{1}{N_{t}}}-1\right)\left(k^{\frac{1}{N_{t}}}-1\right)}\right]\cdot S_{t}$$

Based on the recursive formula and the first item that the individual perception changes with the period, we can see that the individual perception in the epidemic is determined by the memory rate *ρ*, the association rate *k*, the amount of information in each event cycle {*N*_1_, *N*_2_, *N*_3_, …, *N_t_*} and the amount of the single information stimulus {*S*_1_, *S*_2_, *S*_3_, …, *S_t_*}. The simulation process of this model is divided into the following four steps:Variable generation: According to the research assumption, six variables are determined. The independent variables are the time period, T, the memory rate parameter, *ρ*, the association rate parameter, *k*, the information receiving amount for each period, *N* and the amount of the stimulation of the single information for each period, *S* while the dependent variable is the level of the risk perception, *y*. In this case, the time period is a natural number from 1 to *t*, that is, a single row matrix; the size of t depends on the length of the time period studied. The variable assignment code is as follows:

T=20;s=2*ones(1,T);p=0.5;k=0.3; n=10*ones(1,T);y=ones(1,T);

T represents the length of the variable, that is, how many cycles; ones(1,T) represents an element with a value of 1 in the T column of the first row;

2.Formula input: Following the generation of the variables, the first term and the recursive formula of the model are input into the command line window of the MATLAB software, and the variables that have been assigned are used for one operation. When inputting the summation part of the recursive formula of this model, this paper adopts an intermediate variable cycle to circle summation. The model formula code is as follows:

y(1)=s(1)*(p^(1/n(1))*p/(p^(1/n(1))-k^(1/n(1)))/(p^(1/n(1))-1)-k^(1/n(1))*k/(p^(1/n(1))-k^(1/n(1)))/(k^(1/n(1))-1)+1/(p^(1/n(1))-1)/(k^(1/n(1))-1));

for t=2:1:T;

cycle=0;

for i=1:t;

cycle=cycle+k^(1/n(i))/(1-k^(1/n(i)))*k^(t-i);

end

y(t)=p*y(t-1)+s(t)*((p-k)/(p^(1/n(t))-k^(1/n(t)))*(1-k)*(cycle)+(p^(1/n(t))*p/(p^(1/n(t))-k^(1/n(t)))/(p^(1/n(t))-1)-k^(1/n(t))*k/(p^(1/n(t))-k^(1/n(t)))/(k^(1/n(t))-1)+1/(p^(1/n(t))-1)/(k^(1/n(t))-1)));

3.Function generation and extraction: To facilitate the subsequent parameter assignment, the formula is saved as a function in script form, as shown in Figure 1.

The calling function code is as follows:

r1 = perception (0.6, 0.3);

For example, the significance of the variable r1 is that when the memory rate parameter *ρ* is 0.6 and the association rate parameter *k* is 0.3, the value of the risk perception changes with the cycle;

4.Image generation: Using the visualization function of MATLAB, the results are generated.

## 4. Results

### 4.1. Risk Perception Analysis of the Model Parameters

#### 4.1.1. Impact of the Memory Parameters on the Risk Perception

When discussing the influence of the memory parameters *ρ* and *k* on the risk perception, in order to eliminate the influence of other parameters under the assumption of meeting the assumption, we can set the amount of information received by individuals and the amount of single information stimulus in each time period to remain unchanged, that is, *N*_1_ = *N*_2_ = … = *N_t_* = 10, *S*_1_ = *S*_2_ = … = *S_t_* = 2. Furthermore, different combinations of *ρ* and *k* are used to explore their effects, and the combinations of *ρ* and *k* in Table 3 are set and numbered.

In the combination of the above tables, when *ρ* is 0.6, the memory rate of the representative individual is high, and 60% of the previous information can be remembered when receiving the second information. When *ρ* is 0.2, the memory rate of the representative individual is low, and 20% of the previous information can only be remembered when receiving the second information; the variation range of the association rate is 0.2 to 0.4. When *k* equals 0.4, it indicates that the association rate of the individuals is high and the perception of the previous information can be awakened by 40%. When *k* equals 0.2, it means that the perception awakens by 20%.

In order to avoid repeated actions and to make the influence of *ρ* and *k* clearer and intuitive, this paper selects the representative combination ①~⑤ for the simulation.

Figure 2 shows the impact of the five parameters on the individual epidemic perception. With the passage of time, each time period, the individual changes in the level of the epidemic perception. Through the analysis of Figure 2, the following conclusions can be drawn:From the overall situation of the five results, when the amount of information received by the individuals in each period and the amount of the single information stimulus remain unchanged, the individual’s perception level presents a reverse ‘L’ type, which shows that the individual’s perception level increases rapidly at the beginning of the epidemic, then the growth rate of the perception becomes slow and finally remains unchanged at a certain level;By analyzing the results of ①③⑤, namely, under the condition of the constant memory rate *ρ*, with the increase of k, the individual’s perception level will also be significantly improved. Similarly, with the passage of time, this gap will increase in size;By analyzing the results of ②③④, namely, when the association rate *k* = 0.5 and the memory rate *ρ* increases, the individual’s perception level has also improved significantly. Furthermore, as time goes on, the gap in this perception level will increase in size;By analyzing the results of ①② and ④⑤, we find that the changes in the association rate have a significantly greater impact on the risk perception than the changes in the memory rate.

#### 4.1.2. Impact of the Changes in the Individual Information Reception on the Epidemic Perception

In the previous section, we assumed that the total number of information bars N and the single information stimulus S received by the individuals at each time period remained unchanged. In this case, we can obtain that the basic trend of the individual epidemic perception increases first and then remains unchanged. In reality, due to the spread of the epidemic, the government’s prevention and control measures and the occurrence of other events, the public information reception will change. This change is reflected in two aspects, one is the change in the amount of information received, reflecting that the severity of epidemic events will change over time; second, the change of the information stimulus reflects the change of the information content in epidemic events. In this section, in order to discuss the impact of the changes in the individual acceptance of the information and the information stimulus on the epidemic perception, we set the parameters *ρ* = 0.5 and *k* = 0.3 as constant values and then analyze the total amount of information and the amount of the single information stimulus.

Impact of the changes on the total amount of information received

When exploring the trend change of the information reception, the single information stimulus *S*_1_ = *S*_2_ = … = *S_t_* = 2 remains unchanged. In order to more clearly and directly explore the impact of the information change trend on the individual epidemic perception, and according to the change rule of the total amount of information in the disaster information model, this paper will use four simple functions (linear growth function, linear decline function, normal function, and sine function) to simulate the four possible change trends of the total amount of the epidemic information over time: monotonically increasing, monotonically decreasing, first increasing and then decreasing and fluctuation.

A.The total amount of epidemic information increases monotonously over time.

The function of the amount of the individual information that changes with the time period is assumed to be *N_t_* = a*t* + b, and the random generation parameters are a = 5, b = 5. The simulation results of the individual information reception and the individual epidemic perception level are shown in Figure 3.

It can be seen from Figure 3 that when the amount of the individual information received increases linearly, the level of the individual perception also increases, and the growth rate increases.

B.The total amount of the epidemic information decreases monotonously over time.

The function of the amount of the individual information that changes with the time period is assumed to be *N_t_* = a*t* + b, and the random generation parameters a = −5, b = 105. The simulation results of the individual information reception and the individual epidemic perception level are shown in Figure 4.

Figure 4 shows that when the amount of information received by the individual decreases linearly, the individual’s perception level will show first an increase and then a decrease in the trend, and when the perception level reaches the maximum after the fourth time period, then the risk perception level will show a downward trend, and the decline rate gradually decreases.

C.The total amount of the epidemic information increased first and then decreased over time.

When the individual’s information reception shows a trend of increasing first and then decreasing, we use the probability density function of the normal distribution for the simulation analysis, as follows:(3)$$N_{t}=C\cdot \frac{1}{\sqrt{2\pi }\sigma }\cdot \exp\left(-\frac{{\left(t-\mu \right)}^{2}}{2\sigma^{2}}\right)\;\;\;\;\;\;$$

Depending on the actual situation, we should assign the parameters in the formula: *C* = 1000, *μ* = 10, *σ* = 5, and obtain the sequence value of *N_t_* according to the natural number sequence t. The simulation results of the individual information reception and individual epidemic perception level are shown in Figure 5.

It can be seen in Figure 5 that when the individual’s information acceptance showed a trend of first rising and then falling, the change of the individual perception also showed a trend of first rising and then falling. However, we can see that the maximum value of the information acceptance appeared after the tenth time period and the maximum value of the individual epidemic perception level appeared after the eleventh time period. Combined with the decreasing and increasing trend of the information reception, it is reasonable to speculate that the individual epidemic perception may have a “lag phenomenon” relative to the total information reception.

D.The total amount of the epidemic information fluctuated over time.

Finally, the sine function with more intense and regular changes is used as the change of information reception to verify whether the ‘lag phenomenon’ really exists while exploring the law of the individual epidemic perception changes. To be more in line with the actual situation and its needs, we also need to modify the sine function and assign parameters, as follows:(4)$$N_{t}=30sin\left(\frac{\pi }{5}t\right)+50\;\;\;\;\;\;$$

The simulation results of the individual information reception and the individual epidemic perception level are shown in Figure 6.

According to Figure 6, after excluding the proportion of negative/positive information, the “lag phenomenon” exists between the total amount of the positive individual information acceptance and the individual epidemic perception.

2.Effects of the changes in the single information stimulus parameters

In the study of a single information stimulus, the total amount of information received by the individuals is set at 40 per time period. Similarly, in the study of the change of the single information stimulus, we confirm the possible change trend of the information stimulus according to the actual situation and use the corresponding function for the simulation.

A.The amount of the information stimulation increases monotonically over time.

The monotonically increasing amount of the information stimulation over time often occurs in the early stage of the epidemic (incubation period, outbreak period), and the severity and transmission ability of the epidemic are gradually shown in front of the public. At this time, the stimulation of information related to the epidemic to the public is also gradually increasing. We also use the linear increment function for the simulation.
(5)$$S_{t}=ct+d$$
where parameter *c* represents the speed of the change of a single information stimulus over time and parameter *d* represents the starting point of the information stimulus. Considering that changes in the parameters can cause changes in the stimulus *S*, we set several sets of parameters *c*, *d* in Table 4.

At this time, the simulation results of the individual epidemic perception level are shown in Figure 7.

Figure 7 shows the impact of the five parameters of the information stimulus on the individual epidemic perception. Figure 7 shows the following conclusions:From the overall situation, when the amount of the information stimulation shows a monotonous increasing trend, the individual’s epidemic perception level will show an increasing trend, but the growth rate will gradually decline in the previous cycles and will stop falling at a certain node, which is consistent with the growth trend of the information stimulation and it becomes a linear growth;By analyzing the results of ①②③, when the starting point *d* of the information stimulus is constant, the growth rate *c* determines the growth rate of the second half of the perception of epidemic in the middle and late stages of the epidemic event;By analyzing the results of ③④⑤, when the change rate of the information stimulation *c* is constant, the greater the initial information stimulation *d*, the greater the initial growth rate of the level of the epidemic perception in the previous several cycles, and the overall level will be higher in the middle and later periods, which is in line with reality.

B.The amount of the information stimulation decreases monotonically over time.

The situation in which the amount of the information stimulus monotonically decreases with time often occurs in the late stage of the epidemic (resolution period and recovery period). The threat of the epidemic has gradually been lifted and the beneficial information, such as vaccines, has been spread on a large scale. In this case, we can also use the linear incremental function (5) for the simulation.

Considering that the changes in the parameters can cause changes in the stimulus *S*, we set several sets of parameters *c*, *d* in Table 5.

At this time, the simulation results of the individual epidemic perception level are shown in Figure 8.

Figure 8 shows the impact of the five parameters of the information stimulus on the individual epidemic perception. Figure 8 shows the following conclusions:In the general situation, when the amount of the information stimulation shows a monotonic decreasing trend, the individual epidemic perception level will first increase to the peak and then the perception level will begin to decline;By analyzing the results of ①②③, when the starting point *d* of the information stimulus is constant, we can also get the result that the change speed of the single information stimulus c determines the change speed of the latter half of the epidemic perception in the middle and late stages of the epidemic event;By analyzing the results of ③④⑤, we found that the starting point of the information stimulus *d* determines the peak of the individual epidemic perception level and the growth rate of the early epidemic level;By analyzing the overall downward trend, we can find that the perception level of the epidemic may decline to 0 or even become negative in a few cycles, and this situation will not occur in the real epidemic situation because for ①, for the types of epidemic information, although there will be information that will reduce the perception of the epidemic, the amount of such information is quite different from that that can improve the perception level of the epidemic; for ②, when the perception drops to a certain extent, the role of the message that can be the perception of the epidemic will change.

C.The amount of the information stimulation increased firstly and then decreased over time.

The situation in which the amount of the information stimulation increases first and then decreases over time corresponds to the overall change of a wave of epidemics. In addition to the large-scale epidemic that broke out for the first time, the rebound after the case returned to zero, which was similar to the second- and third-wave small-scale epidemics in Beijing, Shanghai, Liaoning and other places, can also be represented by this trend. When using the probability density formula (3) of the normal distribution for the simulation, we also list the assignments, such as in Table 6.

At this time, the simulation results of the individual epidemic perception level are shown in Figure 9.

Figure 9 shows the impact of the five parameters of the information stimulus on the individual epidemic perception. Figure 9 shows the following conclusions:By analyzing the results of ①②⑤⑥, we find that when the overall value of the amount of the information stimulus changes from large to small, the curve of the whole perception level gradually changes from the normal curve to the general perception law curve, that is, the curve that rises first and then remains flat is close. In other words, the larger the amount of the information stimulus, the change law of the public perception level will be more consistent with the change law of the amount of the information stimulus;By analyzing the results of ②③④, we found that when the peak time of the information stimulus changed, the magnitude of the epidemic perception peak did not change.

### 4.2. Simulation Analysis of the Epidemic Risk Perception Process Trend

In this section, we need to determine the key time nodes of the epidemic process according to the hyperbolic crisis management model of COVID-19, analyze the characteristics of the epidemic risk information in stages, assign the two parameters of information quantity and the stimulus quantity by using different functions and finally simulate the perceptual changes of the different populations with multiple parameters of the memory rate and association rate.

#### 4.2.1. Identification of the Life Cycle and the Management Cycle Time Nodes for COVID-19

Standard for the life cycle classification of the COVID-19 epidemic

Based on the theories related to the crisis life cycle management model and the information related to the epidemic situation, we combine the two life cycle classification criteria in Table 1 and Table 2, and summarize the new classification criteria, as shown in Table 7.

In Table 7, most of the classification criteria still follow the classification criteria in Table 1, but there are two problems to be explained. One is the problem of the symptom period and the outbreak period, which are named the symptom period rather than the outbreak period. The reason is that the symptom period includes the previous period of the outbreak period, that is, the epidemic situation does not rapidly become serious in the first half of the symptom period. Second, in the demise period, considering that the imported cases have become the main source of the newly diagnosed cases, the domestic social order has gradually become normal, and the growth of the two clustering cases has not developed into the second wave of national epidemics, indicating that the epidemic situation in China has been controlled, which is consistent with the characteristics of the demise period, which is stable and has potential threats.

2.Standard for the classification of the management cycle of the COVID-19 epidemic

In addition, we summarized the conditions and characteristics of the new coronavirus management cycle, such as Table 8.

3.Summary of nodes

Using the national epidemic data and the relevant news reports, we arrive at a date for the key conditionalities, as shown in Table 9.

By summarizing the life cycle nodes of the epidemic, it can be seen that the latent period of the epidemic lasted 14 days from 8 December 2019 to 25 December 2019; the symptom period lasted 42 days from 26 December 2019 to 5 February 2020; the stalemate (development) period lasted 14 days from 6 February 2020 to 19 February 2020; the decline period lasted 32 days from 20 February 2020 to 23 March 2020 and the extinction period has not ended since it began on 23 March 2020.

The summary of the epidemic management cycle node shows that the recognition period is from 27 December 2019 to 30 December 2019, a total of 4 days; the defense period lasted 5 days from 30 December 2019 to 4 February 2020; the response period lasted 17 days from 4 February 2020 to 21 February 2020; the depletion period lasted 34 days from 21 February 2020 to 27 March 2020, and the rethinking warning period has not ended since it began on 27 March 2020.

#### 4.2.2. Analysis of the Information Characteristics for the Different Periods

Based on the mediating effect of the emotion and information severity, we can deduce that the influence of the information on the public perception. Through an understanding of the characteristics of each stage of the epidemic life cycle and management cycle, we can deduce the changes in the emotion and epidemic severity in the epidemic, and then infer the process of the risk information.

Latent period and the Recognition period (27–30 December 2019)

When the epidemic is in the latent period, the epidemic information has not been sent out, and it is only during the recognition period, that the case is identified, the epidemic risk information will begin to spread. In the recognition period, the risk information is spread only in a small range, and the amount of information and single information are extremely limited for individuals outside of the area that concerns the risk information; for individuals involved in the region, although the amount of information is small, the stimulus for the individuals will be large.

The key node for this period is 30, that is, when Dr. Wenliang Li released an unknown pneumonia information date.

2.Symptom period and the Defense period (31 December 2019 to 4 February 2020)

When the epidemic is in the symptom period and the management is in the defense period, the continuous expansion of the epidemic scale and the gradual strictness of the epidemic prevention measures, after reaching the outbreak point, will improve the level of the public’s risk perception. At this period, the government and media releases risk information, and the amount of information will grow rapidly. The public’s negative emotions account for the vast majority, and the amount of the single information stimulation will gradually increase and reach its peak. At this time, the amount of various rumors and negative news will also gradually increase.

The key node in this period was 20 January 2020. On the one hand, on 19 January, the confirmation of person-to-person transmission was announced. On the other hand, the number of confirmed cases increased rapidly beginning on 20 January, leading to the rapid growth of negative information and information related to epidemics.

3.Stalemate (Development) period and the Response period (4 February to 21 February 2020)

Following the entry into the stalemate period and the response period of the epidemic management, it shows that the epidemic prevention and control measures are effective and the growth rate of the epidemic severity slows down. However, the overall scale is still increasing and the amount of information will remain volatile. The proportion of positive information and the trust of the control measures will increase and the amount of the information stimulation will decrease.

4.Decline period and the Depletion period (21 February to 27 March 2020)

Following the entry into the decline period and the depletion period of the epidemic management, the epidemic has been gradually controlled, and the severity of the epidemic information will also decrease. For the amount of a single information stimulus, the proportion of the gradually relaxed control measures and the positive public emotions will gradually increase, which will lead to the gradual decrease of the information stimulus.

5.Extinction period and the Rethinking Warning period (27 March 2020 to now)

Following the entry into the extinction period of the epidemic and the rethinking of the warning period of the epidemic management, the threat of the domestic epidemic has become recessive. However, due to the growth of the global epidemic and the continuing control measures, the amount of epidemic-related information fluctuates, and the stimulation of the single information will remain volatile due to the fluctuation of the epidemic, and the overall trend is declining.

#### 4.2.3. Parameter Assignment

Period ‘T’

In order to simulate the epidemic from the macroscopic and detailed perspectives, two time periods of T1 and T2 were set. T1 was a 4-day cycle, and the time range was from 27 December 2019 to 2 May 2020. T2 is a cycle of eight days from 27 December 2019 to 12 May 2021.

2.Information stimulus ‘*S*’ and information reception ‘*N*’

According to the analysis of the increase and decrease process of the number of new corona pneumonia infections in the different epidemic periods and the heat change of the related reports, the single information stimulus S and the information reception N were assigned within a reasonable range.

A.In the short period T1, set S1 and N1, as in Table 10.

B.Similarly, in the long period, T2, S2, and N2 were established according to the analysis of the increase and decrease of the number of new corona pneumonia patients in different epidemic periods and the severity change of the related reports, as shown in Table 11.

3Memory rate ‘*ρ*’ and association rate ‘*k*’

For the different individuals, the memory rate and association rate are also different. Four combinations such as Table 12 can be set for the simulation.

#### 4.2.4. Analysis of the Simulation Results

If we want to determine whether the simulation results are effective, we need to measure and obtain the level of the risk perception. This article uses the frequency of the network search behavior to represent the level of the risk perception, which can more easily obtain the trend of the risk perception change.

Simulation Analysis of the Short Period T1

Due to the short period, the influence of the information parameters on the simulation results will be more delicate. In this section, we mainly analyze the influence of the parameter changes.

From Figure 10, Figure 11 and Figure 12, we can draw the following conclusions:The peaks of ①③ appear at the same time and the peaks of ②④ appear at the same time, and the results of ①③ lag behind the results of ②④, but the peak height is far higher than that of the combination of ②④, indicating that individuals with a high memory rate will not only delay the peak of individuals, but will also increase the peak of the risk perception. The higher the peak, the higher the level of the risk perception. That is, individuals with high memory rates have a slower but more sensitive process of awareness of the epidemic during the epidemic;The change of the risk perception in ①② was more tortuous than that in ③④, but the overall perception was still higher than that in ③④, indicating that people with higher association rates would have more obvious perception changes during the outbreak of the epidemic;The lag phenomenon of the information parameters on the perception is still obvious. We can see that the peak of the amount of information received appears in the 14th cycle. The peak of the amount of the information stimulus appears in the 12th cycle. The peak value of ②④ with a low memory rate also appears in the 14th cycle, and the peak value of ①③ with a high memory rate appears in the 16th cycle, indicating that the memory rate of individuals has a greater contribution to the lag effect. We can conclude that the impact of the information reception on the risk perception is higher than that of the information stimulation, because the peak value of the risk perception is closer to the peak value of the information reception.

2.Simulation Analysis of the Long Period T2

When the cycle is longer, the simulation results pay more attention to the overall change of the risk perception.

According to Figure 13, we can see:The evolution process of the risk perception level obtained by the individual perception evolution model is generally consistent with the change process of the user search frequency;Similarly, the effects of the different memory rate parameters and the association rate parameters in the long period are basically the same as those in the short period. The peak time of ①③ is the same, the peak time of ②④ results is the same, and the results of ①③ lag behind the results of ②④, but the peak height is much higher than the combination of ②④; compared with ③④, the changes of the risk perception of ①② are more tortuous, but the overall perception is still higher than that of ③④.

## 5. Discussion

### 5.1. Guidance Strategy for the Public Epidemic Risk Perception Based on the Simulation Results

#### 5.1.1. Targeted Dissemination of the Epidemic Information

This study shows that in the COVID-19 epidemic, the institutional departments related to the risk management must provide the relevant information and knowledge of the epidemic from person to person when releasing information.

The above analysis shows that individuals with higher memory rates are more sensitive to the epidemic information and maintain higher risk perception levels for a long time. Based on this characteristic, it is reasonable to assume that the more educated group will have a higher judgment of the severity of the event after receiving information about the outbreak at the beginning of the outbreak. Because the review and consolidation abilities of the more educated group may be somewhat stronger, they may be more likely to use methods such as replay to improve their memory rate compared with other groups, thus increasing their level of risk perception. For this group, as Vieira [20] said, individuals’ beliefs about risk are related to personal protective behaviors, so we can educate them about risk prevention to make them adopt more personal protective behaviors and thus reduce their risk perception level. Individuals with a higher association rate will have a higher ceiling of the perception level. Consistent with this feature are older groups, especially those who experienced and were greatly affected by SARS in 2003. Having experienced similar situations, the group experienced a higher level of risk for the epidemic. Barbara’s study found that the information frequently disseminated by traditional mass media has a significant positive effect on the public risk perception [21]. Therefore, for such groups, the government should use various information dissemination channels to promote the knowledge about the harm and prevention of the epidemic.

For groups with lower memory and association rates, their judgment of the epidemic is simpler. Once the epidemic situation improves, it is easy to reduce their risk perception to a lower level and then take a more negative risk response. For such groups, the government should take more stringent supervision and, in the bottom line, more resolute epidemic prevention measures.

#### 5.1.2. More Reasonable Information Release Process

In the different stages of the life cycle of the COVID-19 epidemic, the risk management-related institutions should adopt a more stage-specific information release process when releasing the epidemic information.

Latent period and the Recognition period

In the latent period of the epidemic and the recognition period of the management, the government’s task should be to organize the investigation to resolve the hidden dangers of the epidemic, to strive to resolve the epidemic in an unbreakable state, and prepare for an emergency rescue. At this time, the information related to the epidemic should be released from official channels to improve residents’ awareness of prevention.

2.Symptom period and the Defense period

In the symptom period of the epidemic and the management of the defense period, the government has not yet released the information, the sooner the official announcement of the information about the epidemic, the better. Although the public’s risk perception will show an explosive growth in the initial information announcement, the earlier the announcement is made will also allow for the public to take preventive measures earlier to slow down the development of the epidemic.

3.Stalemate (Development) period and the Response period

Following the entry into the stalemate period and the management response period of the epidemic, it is indicated that the epidemic prevention and control measures have played an effective role. At this stage, it is necessary to ensure the disclosure of the information on the epidemic prevention measures and protective effects, and use various methods to detect cases that have not been cured in time.

4.Decline period and the Depletion period/Extinction period and the Rethinking Warning period

Following the entry into the decline period of the epidemic and the depletion period of the management, the epidemic has been gradually controlled, and the public’s risk perception of the epidemic will decrease. At this time, we should pay attention to the information of new cases throughout the country, and publish the information in time to make the local residents pay attention to it, and give early warning of the epidemic in the period of a possible counterattack.

### 5.2. Research Significance

In this study, in the context of the new coronavirus disease, compared with previous studies, the theory of the limited memory is applied to the study of the risk perception in the context of the epidemic. From the perspective of the individual memory, in the study of the impact of the information reception on the individual perception, the parameter of the individual perception stimulus is introduced for the first time, and the previous model is adjusted. Taking into account the law of information release in the context of an epidemic, the difference in the response of the different populations to the epidemic-related information is discussed.

This study also extends to the application of the finite memory theory from the natural disaster risk management to the COVID-19 epidemic scenario. In the model constructed by previous studies and based on the memory theory, the degree of stimulation of a single piece of information for an individual is constant, and the focus of the study tends to be on the memory rate and the association rate parameters, whereas in the COVID-19 epidemic scenario, the progress of the epidemic and major events occur from time to time, which also lead to the uneven release of information. In this study, the model was improved and derived on the basis of the previous studies to enrich the model connotation, so that the model can be better applied in the scenario of public health emergencies, such as the COVID-19 epidemic. In addition, this study not only focuses on the memory rate and association rate parameters, but also explores the effects of the different information release patterns and the types of information on the public perceptions through simulation, which can play an important role in explaining the individual risk coping styles, the formation and evolution of the risk perceptions, and finally the formulation of the relevant strategies.

### 5.3. Limitations and Future Research Perspectives

This study constructed and simulated the risk perception model of the public new coronavirus from the perspective of individual memory, but there are still some limitations in this study:In terms of the model assumptions, this study assumes the public as a complete information audience, that is, the public passively receives all of the information. In fact, the public’s information needs are biased, and the public will change with the evolution of crisis events, which will directly affect the effectiveness of the information release mode. In addition, the truthfulness of the information may also influence the reception of the information by individuals through other dimensions, such as emotions, and thus affect the level of the individual risk perception. In the future, appropriate psychological scales or modeling should be used to explore the mechanisms of the influence of the information truthfulness on the level of the risk perception from that perspective. Meanwhile, this paper lacks consideration for the retelling effect of memory when re-variables are proposed;In terms of a real epidemic simulation, the method of assigning parameters based on the life cycle and the management cycle model of the epidemic is subjective, which will also affect the effectiveness of the simulation results. In terms of verification, although the method using the Baidu search index for the model verification can be used as a reference, more scientific methods are needed to determine the fitting degree of simulation results and indexes;Finally, regarding access to information, this study abbreviated the changes in daily life during the pandemic (e.g., masking, quarantining, social distancing, etc.). This pandemic information cannot be simply obtained from the media as a medium, so there is a need for future in-depth information on this aspect.

## 6. Conclusions

In this paper, the relationship between residents’ epidemic information reception and risk perception in the context of COVID-19 is systematically studied by mathematical modeling. Based on the limited memory theory, the research assumption is proposed, and the evolution model of the public epidemic risk perception is constructed. Then MATLAB software was used for the simulation, and the influence of a series of parameters on the model was obtained, and COVID-19 was simulated and verified. Finally, according to the simulation results, the public epidemic risk perception guidance strategy is proposed. Through simulation and analysis, the main conclusions of this paper are as follows:In the model, we assume that in each time period after the outbreak begins to spread, so has the release of the epidemic-related information, and the amount of information received by the public at each time period is different, and each information received by the public will have different effects on their risk perception. According to the proximity effect, the association effect and the retelling effect, the memory rate parameter and the association rate parameter are set, and the evolution model of the public epidemic risk perception is summarized by a recursive algorithm. The model includes four parameters, namely, memory rate *ρ*, association rate k, information reception N and information stimulation S, in a single period of time;The influence of the different parameters on the risk perception. When the amount of information received N and the information stimulus S remain unchanged, the public’s risk perception is a monotonic upward trend but has an upper limit function, and the upper limit is determined by the memory rate *ρ* and the association rate k, and the influence of the association rate is greater than that of the memory rate. When the amount of information received N and the amount of information stimulus S change, the risk perception will also change, and there is a lag effect, which is determined by the memory rate *ρ*. The impact of the information acceptance on the risk perception is greater than that of the information stimulus.

## Figures and Tables

**Figure 1 ijerph-19-11581-f001:**
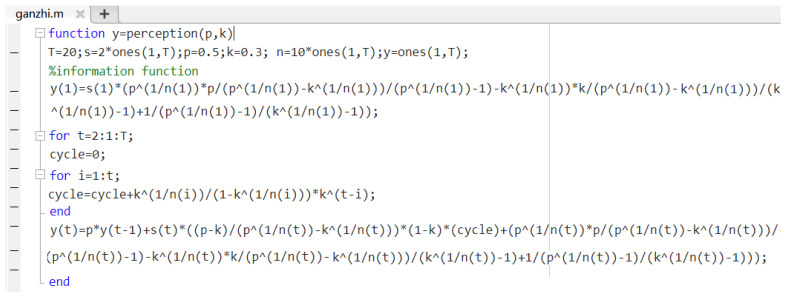
Functions of the risk perception in (p, k).

**Figure 2 ijerph-19-11581-f002:**
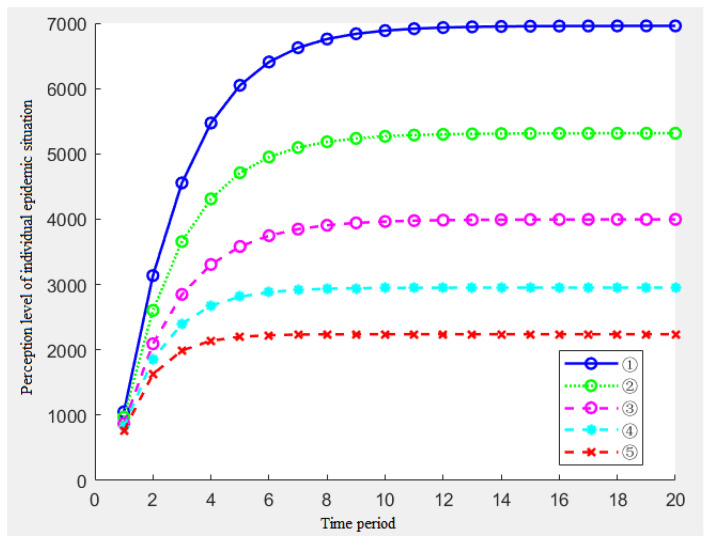
Effects of the five combinations of *ρ* and *k* on the individual epidemic perception.

**Figure 3 ijerph-19-11581-f003:**
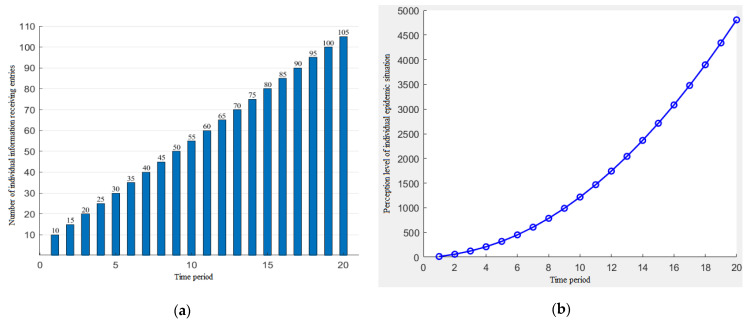
The increasing trend of the individual information reception (**a**) and the change of the individual perception level (**b**).

**Figure 4 ijerph-19-11581-f004:**
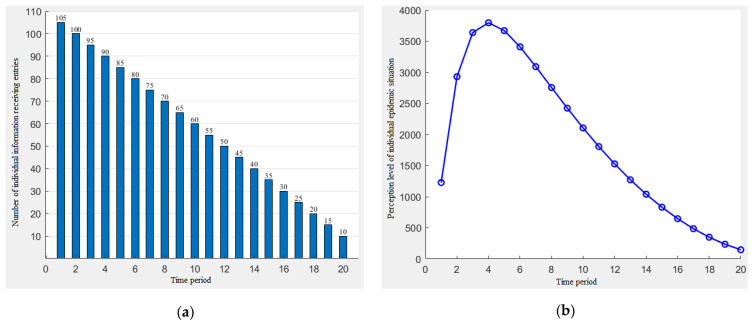
The decreasing trend of the individual information reception (**a**) and the change of the individual perception level (**b**).

**Figure 5 ijerph-19-11581-f005:**
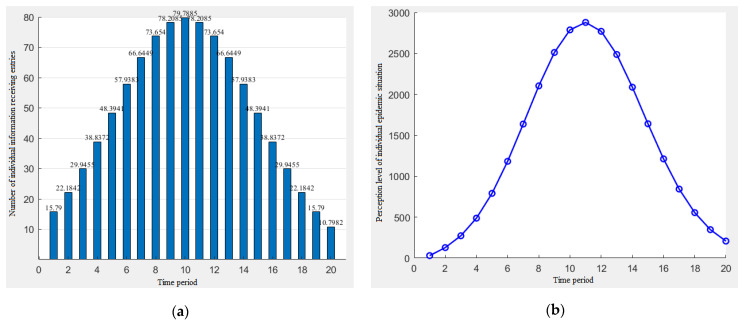
Individual information acceptance increases first and then decreases (**a**) and the change of the individual perception level (**b**).

**Figure 6 ijerph-19-11581-f006:**
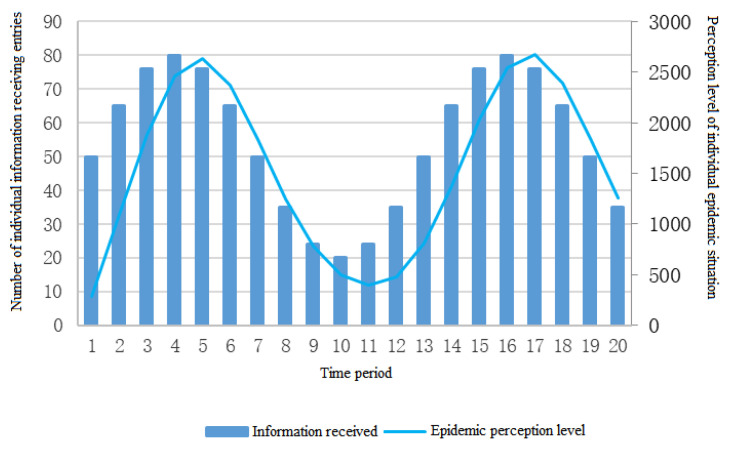
Changes of the risk perception when the individual information receipts fluctuate.

**Figure 7 ijerph-19-11581-f007:**
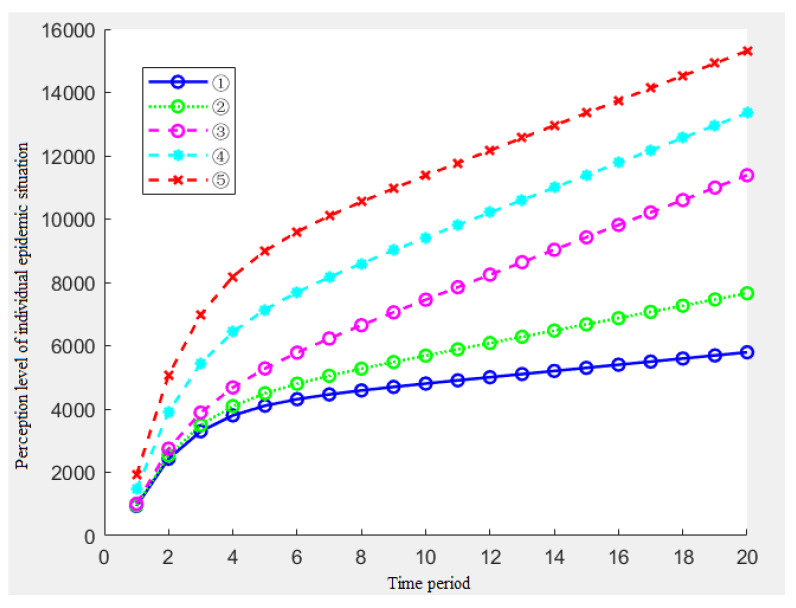
Changes of the epidemic perception when the amount of information stimulus showed an increasing trend.

**Figure 8 ijerph-19-11581-f008:**
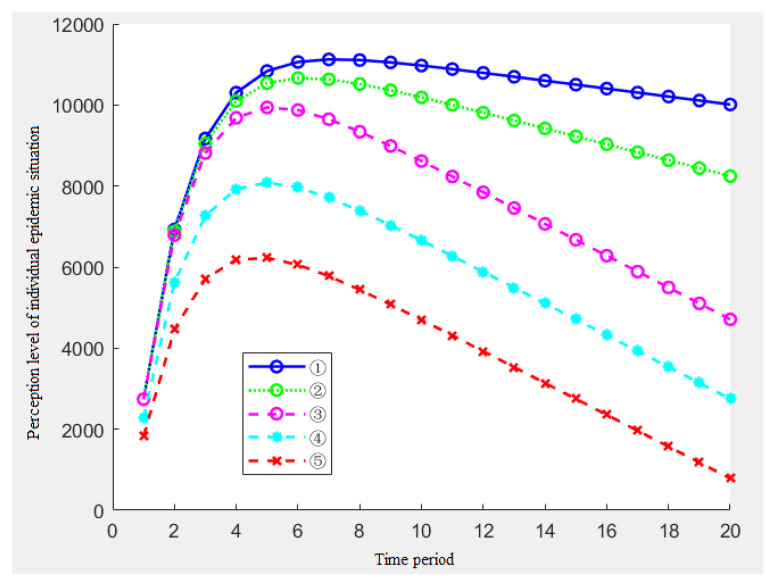
Changes of the epidemic perception when the amount of the information stimulus showed a decreasing trend.

**Figure 9 ijerph-19-11581-f009:**
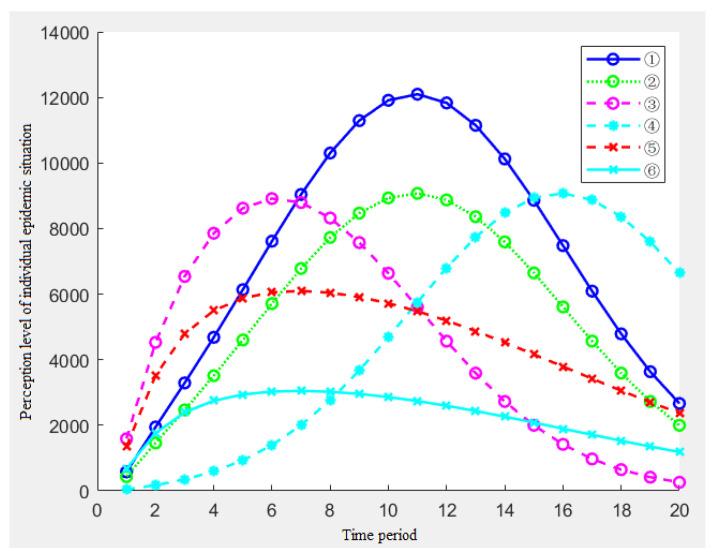
Change of the epidemic perception when the information stimulation first increases and then decreases.

**Figure 10 ijerph-19-11581-f010:**
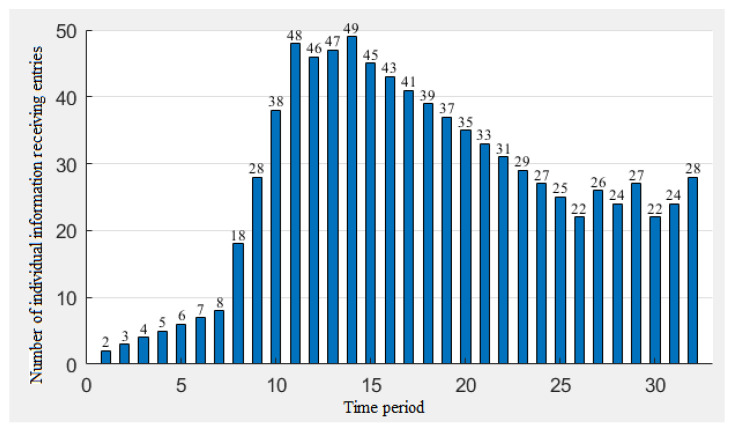
Information reception assignment from 27 December 2019 to 2 May 2020.

**Figure 11 ijerph-19-11581-f011:**
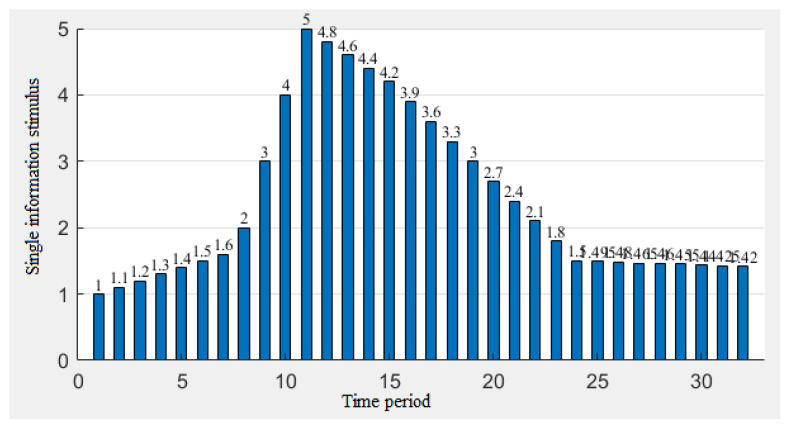
Assignment of the information stimulus from 27 December 2019 to 2 May 2020.

**Figure 12 ijerph-19-11581-f012:**
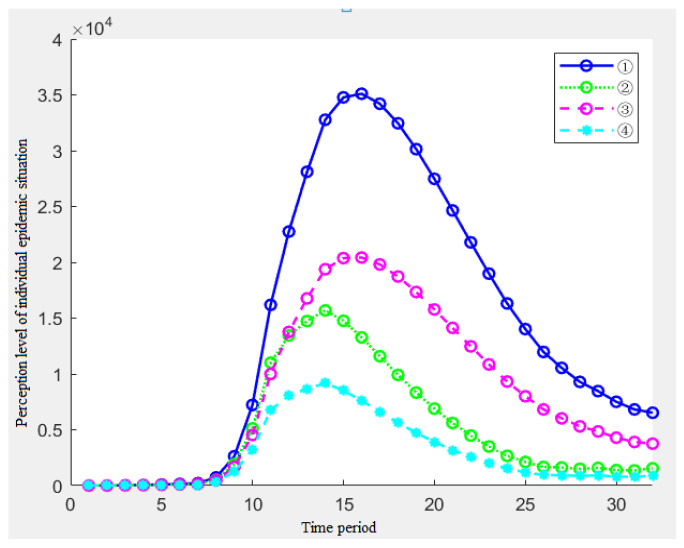
Risk perception changes from 27 December 2019 to 2 May 2020.

**Figure 13 ijerph-19-11581-f013:**
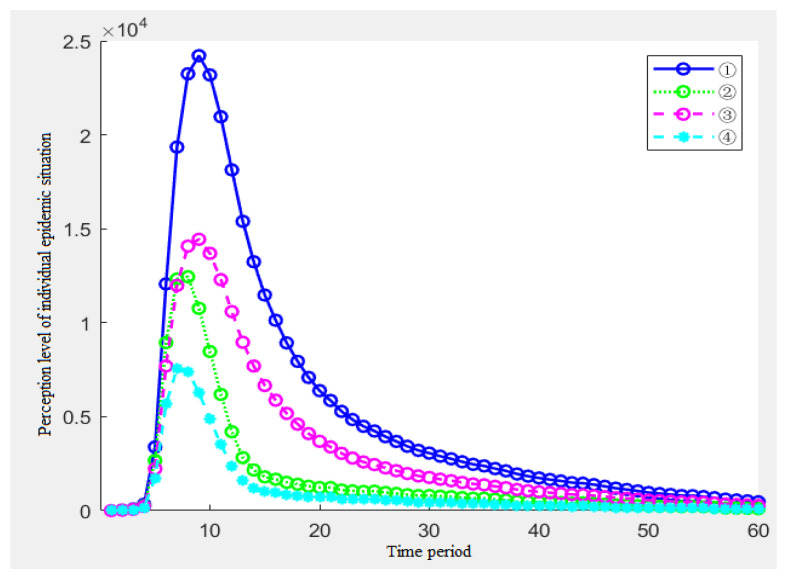
Risk perception changes from 27 December 2019 to 12 May 2021.

**Table 1 ijerph-19-11581-t001:** Critical standard system for the life-cycle stage division of general infectious diseases.

Life Cycle		Condition Index	Trend Index
Latent Period	Beginning	First case of the virus infection	-
	End	First spatial associated group infection	-
Explosive Period	Beginning	First spatial associated group infection	-
	End	Number of newly diagnosed cases on the same day < than on the previous day	The growth trend in the number of newly diagnosed cases has been converted into ups and downs or a slow decline
Stalemate Period	Beginning	Number of newly diagnosed cases on the same day < than on the previous day	The growth trend in the number of newly diagnosed cases has been converted into ups and downs or a slow decline
	End	Number of newly diagnosed cases on the same day < Number of new cases cured that day	The number of newly diagnosed cases decreased and the number of newly cured cases increased
Solution Period	Beginning	Number of newly diagnosed cases on the same day < Number of new cases cured that day	The number of newly diagnosed cases decreased and the number of newly cured cases increased
	End	Number of newly diagnosed cases → 0 and Number of new suspected cases → 0	Maintained state (incidental zero report should not be used as a criterion)
Convalescent Period	Beginning	Number of newly diagnosed cases → 0 and Number of new suspected cases → 0	Maintained state (incidental zero report should not be used as a criterion)
	End	The last patient was cured and discharged	-

Note: The end of the previous phase and the opening of the following phase of the adjacent period are consistent.

**Table 2 ijerph-19-11581-t002:** Standard for the Period Division Based on Emergencies.

Life Cycle		Different Divide Conditions	Characteristics
Latent Period	Beginning	No obvious node	Hidden, asymptomatic, latent crisis exists in the invisible state
	End	Symbolic event occurs	
Symptom Period	Beginning	Symbolic event occurs	Awareness, potential crises triggered by triggers and marked events
	End	A series of joint and several reactions to uncontrolled situations	
Development period	Beginning	A series of joint and several reactions to uncontrolled situations	Highly dominant, continuous deterioration, may be accompanied by other or similar invisible crises. Crisis events spread rapidly and have a wide influence.
	End	Negative growth in the number of affected groups and turning points in the hazard trends	
Decline Period	Beginning	Negative growth in the number of affected groups and turning points in the hazard trends	Significant reductions in the hazard levels, but potential threats remain
	End	No obvious node	
Extinction Period	Beginning	No obvious node	Crisis events were almost completely controlled, social order was basically restored and public life returned to normal
	End	No obvious node	

Note: The end of the previous phase and the opening of the following phase of the adjacent period are consistent.

**Table 3 ijerph-19-11581-t003:** Memory parameter combinations.

	*k*	High Association Rate (0.4)	General Association Rate (0.3)	Low Association Rate (0.2)
*ρ*				
**High memory rate (0.6)**		(0.6, 0.4)	② (0.6, 0.3)	(0.6, 0.2)
**Medium memory rate (0.5)**		① (0.5, 0.4)	③ (0.5, 0.3)	⑤ (0.5, 0.2)
**General memory rate (0.4)**		—	④ (0.4, 0.3)	(0.4, 0.2)
**Low memory rate (0.3)**		—	—	(0.3, 0.2)

**Table 4 ijerph-19-11581-t004:** The assignment of *c*, *d* for the increasing trends.

Number	*c*	*d*	*S* _1_	*S* _20_
①	0.05	2	2.05	3
②	0.1	2	2.1	4
③	0.2	2	2.2	6
④	0.2	3	3.2	7

**Table 5 ijerph-19-11581-t005:** The assignment of *c*, *d* for the decreasing trends.

Number	*c*	*d*	*S* _1_	*S* _20_
①	−0.05	6	5.95	5
②	−0.1	6	5.9	4
③	−0.2	6	5.8	2
④	−0.2	5	4.8	1

**Table 6 ijerph-19-11581-t006:** The assignment of *C*, *μ*, σ under the normal distribution.

Number	*C* (Overall Size)	*μ* (Peak Coordinates)	σ (Rate of Divergence)
①	80	10	5
②	60	10	5
③	60	5	5
④	60	15	5
⑤	80	10	10
⑥	40	10	5

**Table 7 ijerph-19-11581-t007:** New epidemic classification criteria.

Life Cycle		Condition Index	Trend Index
Latent Period	Beginning	First case of the virus infection	-
	End	First spatial associated group infection	-
Symptom Period	Beginning	First spatial associated group infection	-
	End	Number of newly diagnosed cases on the same day < than on the previous day	The growth trend in the number of newly diagnosed cases has been converted into ups and downs or a slow decline.
Stalemate (Development) Period	Beginning	Number of newly diagnosed cases on the same day < than on the previous day	The growth trend in the number of newly diagnosed cases has been converted into ups and downs or a slow decline.
	End	Number of newly diagnosed cases on the same day < Number of new cases cured that day	The number of newly diagnosed cases decreased and the number of newly cured cases increased.
Decline period	Beginning	Number of newly diagnosed cases on the same day < Number of new cases cured that day	The number of newly diagnosed cases decreased and the number of newly cured cases increased.
	End	Number of newly diagnosed local cases → 0 and Number of new suspected local cases → 0	Increasing proportion of the imported cases
Extinction period	Beginning	Number of newly diagnosed local cases → 0 and Number of new suspected local cases → 0	Increasing proportion of the imported cases
	End	End of the global epidemic	-

**Table 8 ijerph-19-11581-t008:** Division conditions and characteristics of COVID-19 management cycle.

Life Cycle	Corresponding Management Cycle	Task and Characteristics
Latent Period	Recognition Period	Timely and accurate identification of the potential risks requires an early intervention, taking coping strategies to eliminate the hidden dangers of the crisis and to avoid emergencies
Symptom Period	Defense Period	Recognizing the crisis, preventing the large-scale outbreak of the crisis, controlling the crisis to a certain extent possible and avoiding contagion
Stalemate (Development) Period	Response Period	Stabilize the situation, try to control the worsening situation of the crisis, and try to stop the worsening trend
Decline Period	Depletion Period	Need to continue to take measures to prevent and control, but also to give the public spiritual comfort, to eliminate negative effects
Extinction Period	Rethinking Warning Period	Summary of the epidemic management

**Table 9 ijerph-19-11581-t009:** Critical time points for the epidemics.

Date	Related Events	Judgment Basis	Representative Node
12 p.m. 8 December 2019	On 11 January 2020, the Wuhan Health Commission issued the ‘Expert Interpretation of the Unexplained Viral Pneumonia Update’, stating that ‘this case of unexplained viral pneumonia in Wuhan occurred between 8 December 2019 and 2 January 2020’.	First case of the virus infection	Latent period began
12 p.m. 25 December 2019	In December, many cases of unexplained pneumonia with an exposure history to the South China Seafood market were found, and on 26 and 30 December, there were two cases of a group diagnosis of unexplained pneumonia.	First Spatially Associated Group Infection	Latent period ended Symptom period began
27 December 2019	Cases of unexplained pneumonia reported by the Hubei Hospital of Integrated Traditional Chinese and Western Medicine to the Wuhan Jianghan district CDC.	Initial confirmation of unexplained pneumonia	Recognition period began
30–31 December 2019	Dr. Wenliang Li explained the information about the unidentified pneumonia. The Wuhan Health Commission issued a ‘briefing on the current situation of pneumonia in our city’ and found 27 cases, prompting the public to take protective measures.		Recognition period ended Defense period began
14–19 January 2020	The national teleconference was held to confirm the characteristics of ‘human transmission’. The epidemic may spread further and the epidemic began to break out.		
23 January 2020	Wuhan channel closed.		
12 p.m. 4 February 2020	There has been an inflection point in the number of new cases of the national epidemic, and an overall downward trend since.	Number of newly diagnosed cases on the same day < Number of newly diagnosed cases on the previous day	Symptom period ended Stalemate (Development) period began
3–5 February 2020	The Central Steering Group has mobilized 22 national emergency medical rescue teams to build shelter hospitals in Wuhan.	Integrated mobilization of national resources	Defense period ended Response period began
12 February 2020	The large increase in the number of new cases on 12 and 13 February was the date of the detection results of the new coronavirus in Wuhan. The number of new cases before and after showed a downward trend, so this node cannot be regarded as the epidemic node.	—	Data showing abnormal nodes
12 p.m. 19 February 2020	From the analysis of the epidemic data, the number of cured cases has significantly exceeded the number of new cases since 19 February, and the confirmed cases have been decreasing since then.	Number of newly diagnosed cases on the day < Number of newly cured cases on the day	Stalemate (Development) period ended Decline period began
21 February 2020	Since the 21st, the provinces have gradually lowered their response levels to the major public health emergencies and gradually lifted the restrictions regarding movement.	Starting to restore social order	Response period ended Depletion period began
12 p.m. 23 March 2020	From the analysis of the epidemic data, the imported cases have become the main newly diagnosed and suspected cases since 24 March, and the locally diagnosed and suspected cases show a floating trend to 0.	Increasing proportion of imported cases	Decline period ended Extinction period began
27 March 2020	Emphasizing the focus on ‘external input, internal rebound’		Depletion period ended Rethinking warning period began
29 April 2020	Imported cases abroad are basically controlled and national epidemic prevention and control are normalized.		

Note: Information on the measures comes from the white paper ‘China action against the new coronavirus pneumonia epidemic’.

**Table 10 ijerph-19-11581-t010:** Assignment to *S* and *N* to the Short Period T1.

Date 1	Period T1	Stimulus S1	Amount of Information N1
27 December 2019~30 December 2019	1	1	2
31 December 2019~3 January 2020	2	1.1	3
4 January 2020~7 January 2020	3	1.2	4
8 January 2020~11 January 2020	4	1.3	5
12 January 2020~15 January 2020	5	1.4	6
16 January 2020~19 January 2020	6	1.5	7
20 January 2020~23 January 2020	7	1.6	8
24 January 2020~27 January 2020	8	2	18
28 January 2020~31 January 2020	9	3	28
1 February 2020~4 February 2020	10	4	38
5 February 2020~8 February 2020	11	5	48
9 February 2020~12 February 2020	12	4.8	46
13 February 2020~16 February 2020	13	4.6	47
17 February 2020~20 February 2020	14	4.4	49
21 February 2020~24 February 2020	15	4.2	45
25 February 2020~28 February 2020	16	3.9	43
29 February 2020~3 March 2020	17	3.6	41
4 March 2020~7 March 2020	18	3.3	39
8 March 2020~11 March 2020	19	3	37
12 March 2020~15 March 2020	20	2.7	35
16 March 2020~19 March 2020	21	2.4	33
20 March 2020~23 March 2020	22	2.1	31
24 March 2020~27 March 2020	23	1.8	29
28 March 2020~31 March 2020	24	1.5	27
1 April 2020~4 April 2020	25	1.495	25
5 April 2020~8 April 2020	26	1.48	22
9 April 2020~12 April 2020	27	1.465	26
13 April 2020~16 April 2020	28	1.46	24
17 April 2020~20 April 2020	29	1.455	27
21 April 2020~24 April 2020	30	1.44	22
25 April 2020~28 April 2020	31	1.425	24
29 April 2020~2 May 2020	32	1.42	28

**Table 11 ijerph-19-11581-t011:** Assignment of *S* and *N* to Long Period T2.

Date 1	Period T1	Stimulus S1	Amount of Information N1
27 December 2019~3 January 2020	1	1.05	3
4 January 2020~11 January 2020	2	1.25	5
12 January 2020~19 January 2020	3	1.45	7
20 January 2020~27 January 2020	4	1.8	13
28 January 2020~4 February 2020	5	3.5	33
5 February 2020~12 February 2020	6	4.9	47
13 February 2020~20 February 2020	7	4.5	48
21 February 2020~28 February 2020	8	4.05	44
29 February 2020~7 March 2020	9	3.45	40
8 March 2020~15 March 2020	10	2.85	36
16 March 2020~23 March 2020	11	2.25	32
24 March 2020~30 March 2020	12	1.65	28
1 April 2020~8 April 2020	13	1.4875	23.5
9 April 2020~16 April 2020	14	1.4625	25
17 April 2020~24 April 2020	15	1.4475	24.5
25 April 2020~2 May 2020	16	1.4225	26
3 May 2020~10 May 2020	17	1.4025	24
11 May 2020~18 May 2020	18	1.3815	24
19 May 2020~26 May 2020	19	1.3605	23
27 May 2020~3 June 2020	20	1.3395	23
4 June 2020~11 June 2020	21	1.3185	24
12 June 2020~19 June 2020	22	1.2975	21
20 June 2020~27 June 2020	23	1.2765	22
28 June 2020~5 July 2020	24	1.2555	22
6 July 2020~13 July 2020	25	1.2345	23
14 July 2020~21 July 2020	26	1.2135	21
22 July 2020~29 July 2020	27	1.1925	21
30 July 2020~6 August 2020	28	1.1715	20
7 August 2020~14 August 2020	29	1.1505	20
15 August 2020~22 August 2020	30	1.1295	21
23 August 2020~30 August 2020	31	1.1085	20
31 August 2020~7 September 2020	32	1.0875	20
8 September 2020~15 September 2020	33	1.0665	19
16 September 2020~23 September 2020	34	1.0455	19
24 September 2020~1 October 2020	35	1.0245	20
2 October 2020~9 October 2020	36	1.0035	18
10 October 2020~17 October 2020	37	0.9825	18
18 October 2020~25 October 2020	38	0.9615	16
26 October 2020~2 November 2020	39	0.9405	17
3 November 2020~10 November 2020	40	0.9195	17
11 November 2020~18 November 2020	41	0.8985	17
19 November 2020~26 November 2020	42	0.8775	17
27 November 2020~4 December 2020	43	0.8565	16
5 December 2020~12 December 2020	44	0.8355	18
13 December 2020~20 December 2020	45	0.8145	16
21 December 2020~28 December 2020	46	0.7935	15
29 December 2020~5 January 2021	47	0.7725	14
6 January 2021~13 January 2021	48	0.7515	15
14 January 2021~21 January 2021	49	0.7305	14
22 January 2021~29 January 2021	50	0.7095	13
30 January 2021~6 February 2021	51	0.6885	14
7 February 2021~14 February 2021	52	0.6675	14
15 February 2021~22 February 2021	53	0.6465	13
23 February 2021~2 March 2021	54	0.6255	16
3 March 2021~10 March 2021	55	0.6045	13
11 March 2021~18 March 2021	56	0.5835	12
19 March 2021~26 March 2021	57	0.5625	10
27 March 2021~3 April 2021	58	0.5415	12
4 April 2021~11 April 2021	59	0.5205	11
12 April 2020~20 April 2020	60	0.4995	10

**Table 12 ijerph-19-11581-t012:** Combinations of the different memory abilities.

	*k*	High Association Rate (0.4)	Low Association Rate (0.2)
*ρ*			
**High memory rate (0.8)**		① (0.8, 0.4)	③ (0.8, 0.2)
**Low memory rate (0.5)**		② (0.5, 0.4)	④ (0.5, 0.2)

## Data Availability

The data presented in this study are available in the Web of Science Core Collection database.

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
