# Peer review of "Evolution Model and Simulation Study of the Public Risk Perception of COVID-19"

_ijerph, 2022, doi:10.3390/ijerph191811581_

Round 1

Reviewer 1 Report

As someone who studies public health emergency preparedness and has an interest in risk perception more generally, I opted to review the manuscript with great interest. Since I am not familiar with the specific analyses used in the study, I am approaching my review from the position of a public health practitioner who might want to use the results of this study to inform their communication strategy during a future emergency. It is in this regard that I believe the most important revisions need to be made to the manuscript. As it currently stands I think it would be very hard for practitioners to assess these findings and implement lessons learned in the field.

A fundamental question I had based on the manuscript is what exactly is the role of memory in risk perception? This really needs to be clarified in the paper, as the paper as written does not convince me of memory’s importance to risk perception. I understand that people's perception of risk shifts as new events occur (rise/fall in cases, rollout of vaccines, etc.) In regard to memory, is it that people forget the level of concern that they previously had and that memory is sparked by new developments?

The relationship between memory and education needs to be clarified in the paper as well.

I was also looking for the manuscript to better explain how the research accounts for the ebbs and flows of information during the pandemic. The manuscript seems to indicate that for purposes of simplifying the model the flow of information kept constant, but that ignores both changes in the flow of information as well as changes in the nature of information (increases in dis/misinformation, contradictory information, etc.)

More detail is needed regarding how the variables used in the model are generated.

Specify what information you think is being forgotten. Even if you weren't paying a lot of  attention to media, a lot of change brought about by the pandemic was enforced by daily life (masking, quarantines, etc.) The manuscript also needs to address that from early on in the pandemic people were filtering new information through different cultural and ideological lenses.

This sentence seems out of place (206-207): This section needs to abstract the public's memory and perception of the epidemic situation and related information.

Proofread for grammatical correctness, clarity, and run-on sentences.

Much is made about how much information about the pandemic people received from social media. Provide citations to support it.

The “retelling effect” needs a clearer definition in the paper.

Provide more background on “individual memory theory” and specify what information you believe the public is ignoring.

I’m not sure what this means:

information closely related to itself (line 63).

On line 37, I believe “workshop rumors” should be “workplace rumors.”

The use of the life cycle model in the manuscript was effective and recommendations regarding communication and messaging were connected well, but the data analysis needs to be better explained so it can act as a bridge between the two.

Thank you for the opportunity to review the manuscript. Good luck with the research in the future.

Author Response

Author Responses to Reviewer’ Comments

The authors would like to thank the reviewer for the constructive comments on our manuscript “Evolution model and simulation study of public risk perception of COVID-19”. The manuscript has been carefully revised following the comments. Please find the detailed responses below.

In particular, the revisions in the revised manuscript are highlighted in red font.

(In the following text, the comments of reviewers are shown in blue and italic font to be distinguished from the authors’ responses.)

Responses to Reviewer #1

As someone who studies public health emergency preparedness and has an interest in risk perception more generally, I opted to review the manuscript with great interest. Since I am not familiar with the specific analyses used in the study, I am approaching my review from the position of a public health practitioner who might want to use the results of this study to inform their communication strategy during a future emergency. It is in this regard that I believe the most important revisions need to be made to the manuscript. As it currently stands I think it would be very hard for practitioners to assess these findings and implement lessons learned in the field.

Response: Thank you for taking your time to read our manuscript. Your positive comments are the driving force of our future research. Please find the explanation and modification we have made as follows.

1-1 A fundamental question I had based on the manuscript is what exactly is the role of memory in risk perception? This really needs to be clarified in the paper, as the paper as written does not convince me of memory’s importance to risk perception. I understand that people's perception of risk shifts as new events occur (rise/fall in cases, rollout of vaccines, etc.) In regard to memory, is it that people forget the level of concern that they previously had and that memory is sparked by new developments?

Response: Thank you for your question. First of all, we answer your initial question: what is the role of memory in risk perception? In the paper, memory rate is used as the independent variable to explore, and there should be some connection and difference between memory and memory rate. We discuss the effect of memory rate ρ, association rate k, information reception N and information stimulation S in a single time period as independent variables on risk perception by developing a model of public risk perception of the new crown outbreak from the perspective of individual memory. In this way, we conclude that "when the memory rate ρ increases, the individual's perception level also increases significantly" (Lines 398~399, Page 10). In addition, when the model is applied to the new epidemic environment, it is concluded that "individuals with high memory rate not only delay the peak of individual risk perception level, but also increase the peak of risk perception"(Lines 717~719, Page 24). The importance of memory itself and risk perception needs to be further explored. In addition, this paper uses the memory rate as the independent variable to explore, there are few studies that have clearly stated whether the memory rate received the influence of new events. The definition of memory rate in this paper is based on the proposed Ebbinghaus forgetting curve. Therefore, this paper is a study of memory rate not being affected by new events. Your suggestion is good, and we will consider using memory or memory rate as the dependent variable for further research afterwards.

1-2 The relationship between memory and education needs to be clarified in the paper as well.

Response: Thank you for your suggestion. Likewise, education may indeed be one of the factors that influence an individual's memory. Different levels of education may cause individuals to repeat the same events differently. According to the Ebbinghaus forgetting curve, the memory rate will keep decreasing if there is no review. Thus, education will be one of the factors that affect memory. However, in this article, the influencing factors of memory are not the focus of the study. Therefore, we think that the content of this part should be added in the future related research.

1-3 I was also looking for the manuscript to better explain how the research accounts for the ebbs and flows of information during the pandemic. The manuscript seems to indicate that for purposes of simplifying the model the flow of information kept constant, but that ignores both changes in the flow of information as well as changes in the nature of information (increases in dis/misinformation, contradictory information, etc.)

Response: Thank you for your suggestion. First, in the paper, to simplify the model, it is considered that each piece of information brings the same perception to the individual in each time period, but the amount of perceptual stimulation by a single piece of information is different in different time periods, and the amount of perceptual stimulation is determined by the composition of the kind of information received by the individual. Thus the total amount of perception brought to the individual by all information in each time period is also different. In addition, the text does consider only the positive and negative information properties, ignoring the true and false information. Therefore, the description of "information is true by default" is added to the paper, and this part is added to the limitations, as follow:

“The authenticity of the information is not considered in this study and the default information is true.” (Lines 290~291, Page 7)

“In addition, the truthfulness of the information also affects the individual's reception of the information and thus the level of individual risk perception. Meanwhile, this paper lacks consideration of the retelling effect of memory when re-variables are proposed.”(Lines 841~844, Page 27)

1-4 More detail is needed regarding how the variables used in the model are generated.

Response: Thanks to your suggestion, we have added more details of the proposed variable process in the text, as follows:

“Based on the previous review of studies related to risk perception influencing factors, it can be seen that there are four broad types of influencing factors for public risk perception: four types of factors: individual public characteristics and experiences, risk nature characteristics, risk information characteristics, and time. ”(Lines 237~240, Page 6)

“The characteristics of individual members of the public include gender, age, occupation, education level, and family cultural background. The differences in life experiences and risk attitudes caused by the characteristics of individual members of the public are the most direct reasons for the differences in individual risk perceptions. Combined with individual memory theory, these individual characteristics tend to influence the individual's memory and association rates. In addition to the characteristics of the individual, the characteristics of the risk itself, as the object of perception, can also have an impact on the risk perception of the risk. In this study, the study context was set during the COVID-19 outbreak (December 27, 2019 to May 2, 2020), and the characteristics of the risk itself can be identified as the characteristics of the COVID-19 epidemic, so its influence is stable and not discussed in the model.”(Lines 241~251, Page 6)

“In addition, for individual perception, the influence of risk information characteristics, such as information source, content, distribution channel, reporting format, and information density, can even exceed the influence of individual characteristics on risk perception. This study will characterize the complexity of information in an epidemic through two variables: information receiving amount for each period and the amount of stimulation of single information for each period.”(Lines 252~257, Page 6)

“Finally, time is an important dimension for the study of the evolutionary process of public risk perception. First, the public will continuously engage in the act of collecting and accumulating information in order to avoid risks and reduce risk perceptions, and second, the information capacity of the public is constant [17], which leads the public to continuously update the information they have, and the utility of the earliest information will continuously diminish in comparison due to the forgetting effect of memory, and the passage of time is a necessary condition for the above process, so time is also an The passage of time is necessary for the above process, so time is also an important influence on risk perception. The specific research hypotheses and variables are presented as follows.”(Lines 258~267, Pages 6~7)

Reference: [17] James, M.C. and P. Allan, Dual Coding Theory and Education. Educational Psychology Review 1991, 3(3).

1-5 Specify what information you think is being forgotten. Even if you weren't paying a lot of  attention to media, a lot of change brought about by the pandemic was enforced by daily life (masking, quarantines, etc.) The manuscript also needs to address that from early on in the pandemic people were filtering new information through different cultural and ideological lenses.

Response: Thank you for your suggestions. The relevant information has been mentioned in this article, but since it is not exhaustive enough, we have made some additions, as follows:

“From the early days of the pandemic, people have been filtering new information through different cultural and ideological lenses.”(Lines 202~203, Page 5)

“Even the less media-focused public can feel the changes (masking, quarantines, etc.) implemented in many daily lives in a pandemic.”(Lines 205~207, Page 5)

1-6 This sentence seems out of place (206-207): This section needs to abstract the public's memory and perception of the epidemic situation and related information.

Response: Thank you for your suggestion, we have removed this sentence.

1-7 Proofread for grammatical correctness, clarity, and run-on sentences.

Response: Thank you for your suggestions, we have re-proofed and re-touched the language of the article to ensure that there are no grammatical errors, incoherent phrases, etc.

1-8 Much is made about how much information about the pandemic people received from social media. Provide citations to support it.

Response: Thank you for your suggestion. We have added relevant descriptions and citations in the text as follows:

“The growing flow of information and rapidly escalating situation has increased the visibility of New Crown Pneumonia in the media and on social media[16].”(Lines 208~210, Page 5)

Reference: [16] Kuperjanov, M., EARLY DAYS OF THE NOVEL CORONAVIRUS: PUBLIC RESPONSE IN SOCIAL MEDIA DURING THE FIRST MONTH OF THE OUTBREAK. FOLKLORE-ELECTRONIC JOURNAL OF FOLKLORE 2021,(82).

1-9 The “retelling effect” needs a clearer definition in the paper.

Response: We have added a more detailed definition of the retelling effect with short distances to provide the reader with a clearer understanding, as follow:

“This means that if we are remembering the same thing, the second time will be more effective than the first. For example, when we remember a phone number, the more times we repeat it, the more firmly we will remember it.”(Lines 128~131, Page 3)

1-10 Provide more background on “individual memory theory” and specify what information you believe the public is ignoring.

Response: Thank you for your suggestion. We have added some background related to individual memory theory, as follows:

“In 1969, Shiffrin and Atkinson proposed a three-stage processing model of memory information[12], which argues that as a complete memory system, it includes sensory memory, short term memory and long term memory. After external information enters the memory system, it goes through these three memory structures for processing, and through the specific function of each memory structure, the memory information also goes through three stages from low to high, and the three memory structures in the three-stage processing model of memory information are independent.”(Lines 111~117, Page 3)

Reference: [12] Shiffrin, R.M. and R.C. Atkinson, Storage and retrieval processes in long-term memory. Psychological Review 1969, 76(2).

1-11 I’m not sure what this means:

 information closely related to itself (line 63).

Response: We've understood your question. Here is a review and elaboration of others' studies. In the text, "information closely related to itself" refers to all information related to the individual, including but not limited to changes in epidemic viruses, changes in medium- and high-risk areas, changes in health care resources, etc., in addition to information on illnesses, cures, and preventive measures.

Reference: [8] Kan S., Hongxia F., Jianmin J., Wendong L., & Zhaoli S., Chinese people's risk perception and psychological behavior of SARS information. Acta Psychology Sinica 2003(04), 546-554. [in Chinese]

1-12 On line 37, I believe “workshop rumors” should be “workplace rumors.”

Response: Thank you for your suggestion.It is true that the wording is wrong here, but it should be “folk rumors”. We have made a correction.

1-13 The use of the life cycle model in the manuscript was effective and recommendations regarding communication and messaging were connected well, but the data analysis needs to be better explained so it can act as a bridge between the two.

Thank you for the opportunity to review the manuscript. Good luck with the research in the future.

Response: Thank you very much for your suggestion. We have made certain modifications to the trend analysis section of the outbreak risk perception process, as follows:

“The higher the peak, the higher the level of risk perception. That is, individuals with high memory rates have a slower but more sensitive process of awareness of the epidemic during the epidemic.”(Lines 719~721, Page 24)

Reviewer 2 Report

Dear authors, I would like to thank the authors for the opportunity to revise the paper entitled Evolution model and simulation study of public risk perception of COVID-19. Based on the limited memory theory and a simulation analysis, the paper aims at investigating the evolution mechanism of the individual risk perception level in regard to the Covid-19 pandemic.

Overall, the paper is well structured by covering each section rigourisly, however, it shows few drawbacks which may undermine its publication at current stage. More specifically, while abstract, introduction and methodology section are detailed, the section showing the theroretical framework seems to be weak as it shows five academic contributions only. Therefore, this section should be definitely strenghtened.

The discussion section should also be improved. More specifically, this section should engage with the existing literature and, based on the results, it should either confirm or disbute what has already been academically found. This section does not engage with the literature, therefore, it does not fully show the contribution of the paper to the current literature.

Last but not least, throughout the paper there are several typos and punctuation errors that should be corrected in order to make the paper flow.

The paper has a great potential, however, it cannot be published at its current status. 

Author Response

Author Responses to Reviewer’ Comments

The authors would like to thank the reviewer for the constructive comments on our manuscript “Evolution model and simulation study of public risk perception of COVID-19”. The manuscript has been carefully revised following the comments. Please find the detailed responses below.

In particular, the revisions in the revised manuscript are highlighted in red font.

(In the following text, the comments of reviewers are shown in blue and italic font to be distinguished from the authors’ responses.)

Responses to Reviewer #2

Dear authors, I would like to thank the authors for the opportunity to revise the paper entitled Evolution model and simulation study of public risk perception of COVID-19. Based on the limited memory theory and a simulation analysis, the paper aims at investigating the evolution mechanism of the individual risk perception level in regard to the Covid-19 pandemic.

Response: Thank you for taking your time to read our manuscript. Your positive comments are the driving force of our future research. Please find the explanation and modification we have made as follows.

2-1 Overall, the paper is well structured by covering each section rigourisly, however, it shows few drawbacks which may undermine its publication at current stage. More specifically, while abstract, introduction and methodology section are detailed, the section showing the theroretical framework seems to be weak as it shows five academic contributions only. Therefore, this section should be definitely strenghtened.

Response: First of all, thank you very much for your recognition and affirmation of this study. Secondly, in terms of the theoretical framework, this paper mainly relies on finite memory theory to construct a model of the evolution of the level of public risk perception, and the variables are built up to be interrelated with risk perception according to the theory. The presentation of the theoretical content may be a bit thin, so we add some notes about the theoretical support part to supplement it, as follow. In the future, we will conduct related research based on richer theories in future studies.

“In 1969, Shiffrin and Atkinson proposed a three-stage processing model of memory information[12], which argues that as a complete memory system, it includes sensory memory, short term memory and long term memory. After external information enters the memory system, it goes through these three memory structures for processing, and through the specific function of each memory structure, the memory information also goes through three stages from low to high, and the three memory structures in the three-stage processing model of memory information are independent.”(Lines 111~117, Page 3)

Our definition of the recapitulation effect of memory was not described clearly enough, so we added some notes, as follow.

“Sendhil [13] summed up the association effect of memory (the association effect of memory, also known as the logical miscalculation effect, refers to the association being the psychological process of one thing to think of another thing. This means that if we are remembering the same thing, the second time will be more effective than the first. For example, when we remember a phone number, the more times we repeat it, the more firmly we will remember it.”(Lines 126~131, Page 3)

2-2 The discussion section should also be improved. More specifically, this section should engage with the existing literature and, based on the results, it should either confirm or disbute what has already been academically found. This section does not engage with the literature, therefore, it does not fully show the contribution of the paper to the current literature.

Response: Thank you very much for your suggestion. We apologize for overlooking the comparative and dialectical aspects of previous studies in the discussion section, for which we have added and revised accordingly, as follows:

“For this group, as Vieira[20] said, individuals' beliefs about risk are related to personal protective behaviors, so we can educate them about risk prevention to make them adopt more personal protective behaviors and thus reduce their risk perception level.”(Lines 763~766, Pages 25~26)

“Barbara's study found that information frequently disseminated by traditional mass media has a significant positive effect on public risk perception[21].”(Lines 770~771, Page 26)

Reference:

[20] Vieira, K.M., et al., A Pandemic Risk Perception Scale. Risk analysis : an official publication of the Society for Risk Analysis 2021, 42(1).

[21] Barbara, C. and S. Paul, Newspaper Coverage of Causes of Death. Journalism & Mass Communication Quarterly 1979, 56(4)

In addition, we mention the innovative nature of the study in the paper. The paper compares with previous studies and draws conclusions based on this study. The description of this section is not clear enough, so we have modified it accordingly as follows.

“This study also extends the application of finite memory theory from natural disaster risk management to the COVID-19 epidemic scenario. In the model constructed by previous studies based on memory theory, the degree of stimulation of a single piece of information for an individual is constant, and the focus of the study tends to be on memory rate and association rate parameters, whereas in the COVID-19 epidemic scenario, the progress of the epidemic and major events occur from time to time, which also lead to uneven release of information. In this study, the model was improved and derived on the basis of previous studies to enrich the model connotation, so that the model can be better applied in the scenario of public health emergencies such as the COVID-19 epidemic. In addition, this study not only focuses on memory rate and association rate parameters, but also explores the effects of different information release patterns and types of information on public perceptions through simulation, which can play an important role in explaining individual risk coping styles, the formation and evolution of risk perceptions, and finally the formulation of relevant strategies.”(Lines 819~832, Page 27)

2-3 Last but not least, throughout the paper there are several typos and punctuation errors that should be corrected in order to make the paper flow.

The paper has a great potential, however, it cannot be published at its current status.

Response: Thank you for your suggestions, we have re-proofed and re-touched the language of the article to ensure that there are no grammatical errors, incoherent phrases, etc.

Reviewer 3 Report

The subject of this research undertaken during the COVID19 pandemic is extremely important. The reviewer took note of the obtained results with great care. Since the work is published in English, linguistic corrections should be made in many places. Below, the reviewer indicated the most important remarks.

1.       In line 224 there should be N_t instead of Nt (subscript t)

2.       In line 231, instead of Si, there should be S_i (subscript i)

3.       In line 247, there is part of the equation that is incomplete: M(t) = ???

4.       In formula (2), Y_N_t and Y_N_t-1 should be instead of Y_N (t) and Y_N (t-1), respectively.

5.       In line 278, a fragment of the script contained in Figures 1 has been duplicated. Instead, please describe the rationale for the adopted values and quote the definition of the MATLAB ones function.

6.       A similar comment applies to the remaining points, ie lines 285-295.

7.       The work is in English, so instead of "xunhuan" please use a variable name in English, eg cycle, loop, sum, etc.

8.       Similarly, instead of the function name "ganzhi", please use an English equivalent, e.g. perception, RiskPerception or PerceptionLevel, etc.

9.       In the formula in lines 373 and 383, instead of N_t = at + b, please correct for N_t = a_t + b (t subscript). Also, in dependence (5), instead of S_t = ct + d, there should be S_t = c_t + d. Formula (5) appears many times in the paper, ie after lines 435 and 466. Redundant definitions should be deleted or the formula corrected and the others renumbered. Lines 467-469 are copies of lines 436-439.

10.   Are the values c_1 - c_19 not missing in tables 4 and 5? Should the headings in tables 10 and 11 not be different?

11.   The work presents three research questions contained in lines 95-101. At the end of the thesis, there should be a synthetic overview of the answers to these questions. Research hypotheses can be marked as H1-H3 and properly referred to in the paper.

Author Response

Author Responses to Reviewer’ Comments

The authors would like to thank the reviewer for the constructive comments on our manuscript “Evolution model and simulation study of public risk perception of COVID-19”. The manuscript has been carefully revised following the comments. Please find the detailed responses below.

In particular, the revisions in the revised manuscript are highlighted in red font.

(In the following text, the comments of reviewers are shown in blue and italic font to be distinguished from the authors’ responses.)

Responses to Reviewer #3

The subject of this research undertaken during the COVID19 pandemic is extremely important. The reviewer took note of the obtained results with great care. Since the work is published in English, linguistic corrections should be made in many places. Below, the reviewer indicated the most important remarks.

Response: Thank you for taking your time to read our manuscript. Your positive comments are the driving force of our future research. Please find the explanation and modification we have made as follows.

3-1 In line 224 there should be N_t instead of Nt (subscript t)

In line 231, instead of Si, there should be S_i (subscript i)

In line 247, there is part of the equation that is incomplete: M(t) = ???

In formula (2), Y_N_t and Y_N_t-1 should be instead of Y_N (t) and Y_N (t-1), respectively.

Response: Thank you for your suggestion. We apologize for some incorrect text formatting in the text, we have made corrections and additions.

3-2 In line 278, a fragment of the script contained in Figures 1 has been duplicated. Instead, please describe the rationale for the adopted values and quote the definition of the MATLAB ones function.

 A similar comment applies to the remaining points, ie lines 285-295.

Response: Thank you very much for your advice.

First of all, the assignment of the variables is subjective and based on the development of the epidemic and the characteristics of the epidemic change. We also mentioned this in the limitations section of the article and hope to optimize it in future studies.

Secondly, we have explicitly labeled the specific meaning of the ones function in MATLAB(Lines 332~333, Page 8): ones(1,T) represents an element with a value of 1 in the T column of the first row.

Thirdly, as we mentioned in the text, we use the script form to save the formula as a function in order to facilitate the parameter assignment afterwards. Therefore, the content of the function shown in Figure 1 will have recurring formulas.(Lines 350~351, Page 8)

3-3  The work is in English, so instead of "xunhuan" please use a variable name in English, eg cycle, loop, sum, etc.

Similarly, instead of the function name "ganzhi", please use an English equivalent, e.g. perception, RiskPerception or PerceptionLevel, etc.

Response: Thank you for your suggestion. We have changed the names of the intermediate variables and functions used in the model as you suggested, as follow.

“xunhuan”→“cycle”, “ganzhi”→“perception”

3-4 In the formula in lines 373 and 383, instead of N_t = at + b, please correct for N_t = a_t + b (t subscript). Also, in dependence (5), instead of S_t = ct + d, there should be S_t = c_t + d. Formula (5) appears many times in the paper, ie after lines 435 and 466. Redundant definitions should be deleted or the formula corrected and the others renumbered. Lines 467-469 are copies of lines 436-439.

Response: Thank you very much for your advice.

First, the t in the two equations you mentioned is not a subscript but a variable in the equation. at+b and ct+d both express the linear relationship between the amount of individual information received and the amount of a single information stimulus versus time.

Secondly, we have taken your suggestion and removed the duplicate formula definitions, and marked and explained them.

3-5 Are the values c_1 - c_19 not missing in tables 4 and 5? Should the headings in tables 10 and 11 not be different?

Response: First, as in the previous problem, c in the formula represents the rate of change in the amount of a single information stimulus over time. We make four different assignments to c to reflect the different magnitudes of change, and there is no c1~c20.

Second, the headings of Table 10 and Table 11 are different. We set two different periods in order to be able to simulate the epidemic from both macroscopic and detailed perspectives. Table 10 is the short period with a 4-day cycle, and Table 11 is the long period with an 8-day cycle, which are very different.

3-6 The work presents three research questions contained in lines 95-101. At the end of the thesis, there should be a synthetic overview of the answers to these questions. Research hypotheses can be marked as H1-H3 and properly referred to in the paper.

Response: Thank you for your suggestion. Your proposal is very reasonable, and we did raise three questions in the paper, but the relationship between the hypotheses and these three questions conducted in this study before building the model is intersecting. That is, there is not a complete one-to-one correspondence, so the hypotheses are not presented in the article in the form of H1~H3. We will pay attention to this point in our future research.

Round 2

Reviewer 1 Report

In regard to point 1-2, since the issue of education is raised in the paper, I think some clarification, even if minor, is merited. 

In regard to point 1-3, addressing the role of information in regard to risk perception in terms of COVID-19 without considering the role of mis/disinformation seems like a significant weakness. You should at least indicate how the role of mis/disinformation could be addressed using similar methodology in the future. 

For point 1-4, the clarification is appreciated but I would still like to know from where the information, particularly about individual members of the public, was derived. Where did you originally obtain that information?

For point 1-5, my original was is it information about masking, quarantining, social distancing, that is being forgotten?

For point 1-8, I was looking for evidence that social media was the predominant source for people's information regarding the pandemic. 
